# Transient modeling of the ground thermal conditions using satellite data in the Lena River Delta, Siberia

Sebastian Westermann[1], Maria Peter[1,2], Moritz Langer[3,2], Georg Schwamborn[2], Lutz Schirrmeister[2], Bernd Etzelmüller[1], and Julia Boike[2]

[1]Department of Geosciences, University of Oslo, P.O. Box 1047, Blindern, 0316 Oslo, Norway
[2]Alfred Wegener Institute Helmholtz Center for Polar and Marine Research, Telegrafenberg A43, 14473 Potsdam, Germany
[3]Department of Geography, Humboldt-University, Unter den Linden 6, 10099 Berlin, Germany

*Correspondence to:* Sebastian Westermann
(sebastian.westermann@geo.uio.no)

**Abstract.** Permafrost is a sensitive element of the cryosphere, but operational monitoring of the ground thermal conditions on large spatial scales is still lacking. Here, we demonstrate a remote-sensing based scheme that is capable of estimating the transient evolution of ground temperatures and active layer thickness by means of the ground thermal model CryoGrid 2. The scheme is applied to an area of approx. $16\,000\,\text{km}^2$ in the Lena River Delta in NE Siberia for a period of 14 years.

The forcing data sets at $1\,\text{km}$ spatial and weekly temporal resolution are synthesized from satellite products (MODIS Land Surface Temperature, MODIS Snow Extent, GlobSnow Snow Water Equivalent) and fields of meteorological variables from the ERA-interim reanalysis. To assign spatially distributed ground thermal properties, a stratigraphic classification based on geomorphological observations and mapping is constructed which accounts for the large-scale patterns of sediment types, ground ice and surface properties in the Lena River Delta.

A comparison of the model forcing to in-situ measurements on Samoylov Island in the southern part of the study area yields an acceptable agreement for the purpose of ground thermal modeling, both for surface temperature, snow depth and timing of the onset and termination of the winter snow cover. The model results are compared to observations of ground temperatures and thaw depths at nine sites in in the Lena River Delta suggesting that thaw depths are in most cases reproduced to within $0.1\,\text{m}$ or less and multi-year averages of ground temperatures within 1 to $2°\text{C}$. Comparison of monthly average temperatures at

depths of 2 to $3\,\text{m}$ in five boreholes yielded an RMSE of $1.1°\text{C}$ and a bias of $-0.9°\text{C}$ for the model results. The highest ground temperatures are calculated for grid cells close to the main river channels in the south, as well as areas with sandy sediments and low organic and ice contents in the central delta, where also the largest thaw depths occur. On the other hand, the lowest temperatures are modeled for the eastern part, an area with low surface temperatures and snow depths. The lowest thaw depths are modeled for Yedoma permafrost featuring very high ground ice and soil organic contents in the southern parts of the delta.

The comparison to in-situ observations indicates that transient ground temperature modeling forced by remote sensing data is generally capable of estimating the thermal state of permafrost and its time evolution in the Lena River Delta. The approach could hence be a first step towards remote detection of ground thermal conditions and active layer thickness in permafrost areas.

# 1 Introduction

Permafrost is an important element of the terrestrial cryosphere, which is likely to undergo major transformations in a warming climate in the 21st century. At present, near-surface permafrost covers about a quarter of the land area of the Northern Hemisphere, but future projections with Earth System Models (ESMs) suggest a reduction between 30 and 70% until 2100, depending on the applied anthropogenic emission scenario (e.g. Lawrence et al., 2012). Observations of the ground thermal state are evidence that the ground is already warming in many permafrost areas (Romanovsky et al., 2010) and near-surface permafrost is in the process of disappearing from peripheral areas (e.g. Borge et al., 2016). In-situ monitoring efforts are coordinated world-wide within the Global Terrestrial Network for Permafrost (GTN-P, www.gtnp.org, Burgess et al., 2000) which is comprised of two components: (1) the Circumpolar Active Layer Monitoring (CALM) with measurements of active layer thickness at about 250 sites, and (2) the Thermal State of Permafrost (TSP) in which ground temperatures are measured in over 1000 boreholes with depths ranging from a few to more than $100\,\mathrm{m}$.

While GTN-P can deliver high-quality direct observations of permafrost state variables, TSP and CALM sites represent point measurements on spatial scales of $100\,\mathrm{m}$ and less. Transferring this knowledge to larger regions is hampered by the considerable spatial variability of the ground thermal regime (which limits the representativeness of a measurement) and the strong concentration of TSP and CALM sites in a few regions, while vast permafrost areas are not at all covered (Biskaborn et al., 2015).

A possibility to infer ground temperatures on large spatial scales is the use of grid-based models that use meteorological data as forcing. Spatially distributed permafrost modeling was e.g. demonstrated by Zhang et al. (2013) and Westermann et al. (2013) forced by interpolations of meteorological measurements, or by Jafarov et al. (2012) and Fiddes et al. (2015) by downscaled atmospheric model data. Remote sensing data sets have been extensively used to indirectly infer the ground thermal state through surface observations, e.g. occurrence and evolution of thermokarst features (e.g. Jones et al., 2011), vegetation types characteristic for permafrost (Panda et al., 2014), or change detection of spectral indices (Nitze and Grosse, 2016). As permafrost is a subsurface temperature phenomenon, it is not possible to observe it directly from satellite-borne sensors. However, remotely sensed data sets can be used as input for the above-mentioned permafrost models (Hachem et al., 2009; Westermann et al., 2015).

Langer et al. (2013) demonstrated and evaluated a transient ground temperature modeling scheme forced by remote sensing data for a point in the Lena River Delta. In this work, we update and extend this earlier approach to facilitate spatially distributed mapping of the ground thermal regime based on satellite-derived data sets on surface temperature and snow cover. The model results are compared to in-situ observations of ground temperatures and thaw depths, thus facilitating a coarse assessment of the performance of the scheme regarding important permafrost variables.

## 2 Study area

### 2.1 The Lena River Delta

The Lena River Delta (LRD) is located in NE Siberia at the coast of the Laptev Sea. It constitutes one of the largest river deltas in the Arctic, covering an area of around $32\,000\,\text{km}^2$ between 72 and 74°N. The LRD is dominated by continuous permafrost in a continental climate, with extremely cold winter and relatively warm summer temperatures (Boike et al., 2013). Mean annual ground temperatures are the order of -10 °C, and the frozen ground is estimated to extend to about 400 to 600 m below the surface (Yershov et al., 1991).

With elevations between 0 and 60 m a.s.l., the LRD can essentially be regarded as "flat", so that medium and low resolution data sets (1 km or coarser) can be employed without the need of topographic corrections. However, the surface and ground properties feature a strong heterogeneity at spatial scales of 1 m to 1 km (with e.g. a large number of small water bodies, Muster et al., 2012, 2013) that is not reflected in medium and low resolution data sets. Despite such small-scale variability, the LRD can be classified in three main geomorphological units (Fig. 1), which have distinctly different characteristics regarding their surface and subsurface properties, such as ground ice contents, thermokarst features and vegetation cover (Morgenstern et al., 2013; Fedorova et al., 2015).

The *first river terrace* covers large parts of the eastern and central delta. It is the youngest and most active part of the delta, shaped by river erosion and sedimentation during the Holocene. Polygonal tundra with mosses, sedges, grass and occasional dwarf shrubs dominates the surface (Schneider et al., 2009; Boike et al., 2013). The subsurface material consists of silty sands and organic matter in alluvial peat layers with thicknesses up to 5 to 6 m (Schwamborn et al., 2002b). Ice wedges of more than 9 m depth have been described on the first terrace (Grigoriev et al., 1996; Schwamborn et al., 2002b). The ice contents in the uppermost few meters reach 60 to 80% in volume, while the mineral and organic contents reach 20-40% and 5-10%, respectively (Kutzbach et al., 2004; Zubrzycki et al., 2012). A considerable fraction of the first terrace is composed of the modern floodplain of the Lena River which is periodically inundated. These floodplain areas feature a different ground stratigraphy, with sandy, generally well-drained soils with low organic contents.

The *second river terrace*, located in the northwestern part of the LRD, was created by fluvial deposits between 30 and 15 kaBP when the sea level was lower than today. These sandy sediments generally feature low ice and organic contents (Schirrmeister et al., 2011). Arga Island is the biggest island of this terrace and the geomorphologic unit is often called Arga complex.

The *third river terrace* is composed of late Pleistocene sediments which have not been eroded by the Lena River during the Holocene. It is distributed in isolated islands in the southern margins of the LRD (Grigoriev, 1993; Zubrzycki et al., 2012). The third terrace is part of the Yedoma region which contains substantial quantities of ground ice and organic carbon down to several tens of meters below the surface (Strauss et al., 2013). The Yedoma was accumulated during the extremely cold climate of the last glacial period between 43 and 14 ka and contains ice wedges of more than 25 m depth (Grigoriev, 1993; Schwamborn et al., 2002b; Schirrmeister et al., 2003). The vegetation consists of thick 0.1 to 0.2 m hummocky grass, sedge and moss cover, and the upper horizon of the soil has a thick organic layer. Holocene permafrost degradation resulted in the

current complex thermokarst landscape characterized by thermokarst lakes and drained basins (Morgenstern et al., 2013). The three river terraces occur in clusters of at least a few square kilometers (Fig. 1) so that they can be resolved by grid-based mapping at 1 km scale. A model study by Westermann et al. (2016) suggests that the subsurface stratigraphies of the three river terraces lead to a distinctly different ground thermal regime and susceptibility to future surface warming. Spatially distributed permafrost modeling hence must account for these geomorphological units and their characteristics of subsurface heat transfer.

## 2.2 Field sites and in-situ observations

### 2.2.1 The Samoylov Permafrost Observatory

Samoylov Island is an about four square kilometer large island (72°22'N, 126°28'E) located at the southern apex of the LRD, close to where the the Olenyokskaya Channel flows out of the main stem of the Lena River (Fig. 1). It is situated on the first river terrace and dominated by wet polygonal tundra and thermokarst lakes and ponds of various sizes (Boike et al., 2013). A Russian-German research station has been operating on Samoylov Island for more two decades and facilitated scientific studies on energy and carbon cycling (Kutzbach et al., 2007; Wille et al., 2008; Sachs et al., 2010; Abnizova et al., 2012, e.g.), validation of satellite data sets (Langer et al., 2010) and ESM development (e.g Ekici et al., 2014; Yi et al., 2014; Chadburn et al., 2015). Permafrost temperatures have been increasing, and ice-wedge degradation is occurring "subtly" on sub-decadal timescales, but with long term consequences for the hydrologic drainage (Liljedahl et al., 2016). A detailed overview on the climate, permafrost, vegetation, and soil characteristics on Samoylov Island is provided by Boike et al. (2013). On Samoylov Island, a long time series of meteorological and environmental variables is available (Boike et al., 2013) and forms an excellent basis for validation of satellite data sets and ground thermal modeling (Langer et al., 2010, 2013; Westermann et al., 2016). In the following, we briefly describe the in-situ data sets employed in this study (Sects. 4.1.1 and 4.2.1):

*Surface temperature*: On Samoylov Island, surface (skin) temperature has been measured continuously since 2002 by a downward facing long wave radiation sensor (CG1, Kipp & Zonen, Netherlands). The outgoing long wave radiation is converted to surface temperature using the Stefan-Boltzmann law (see Langer et al., 2013, for details).

*Snow depth and properties:* On the point scale, snow depth measurements have been conducted with an ultra-sonic ranging sensor (SR50, Campbell Scientific, USA; located close to the long wave radiation sensor) since summer 2003, but a few winter seasons are not covered due to sensor failure. In addition, a spatially distributed survey of snow depths and densities (216 points in polygonal tundra) was conducted in early spring 2008 (25 April to 2 May) before the onset of snowmelt (Boike et al., 2013). The onset and termination of the snow cover were manually determined from pictures taken by an automated camera system, with dates from 1998 to 2011 provided in Boike et al. (2013).

*Ground temperature:* In this study, we make use of measurements of active layer temperatures in a low-center polygon established in 2002, and ground temperatures in a 26 m deep borehole since 2006 (Boike et al., 2013). The measurement site of the active layer temperatures can be considered representative for the polygonal tundra of the first river terrace (Boike et al., 2013). The deep borehole is located near the southern bank of the island close to the research station in an area with ground properties that differ from the "typical" stratigraphy of the first terrace: the area around the borehole features sandier soils

with low organic contents that are generally well-drained due to the proximity to the river bank. In the course of an upgrade of the research station, new buildings and structures were erected in the direct vicinity of the borehole in summer 2012 (See Supplementary Material), leading to much higher snow accumulation around the borehole in the following winters (compared to the surrounding terrain on Samoylov Island). Therefore, only borehole data until summer 2012 are used for comparison to model results.

*Thaw depth:* Oriented at the measurement protocol for CALM sites (Burgess et al., 2000), thaw depths have been manually mapped on a grid with 150 points in polygonal tundra on Samoylov Island since 2002. According to the land cover classification in Boike et al. (2013), the grid points are located both on dry polygon rims and wet polygon centers. In most years, several surveys are available covering the entire period from the onset of thaw until maximum thaw depths are reached.

### 2.2.2   In-situ observations in the LRD

Outside of Samoylov Island, only sparse observations on the ground thermal regime are available. In 2009 and 2010, ground temperature measurements at several meters depth were established in four boreholes distributed across the LRD (Fig. 1), all of which are located in a rather homogeneous surroundings (see Supplementary Material for images):

– Olenyokskaya Channel, mouth: located on the third terrace at the W edge close to the Laptev Sea (72°49'20.1" N, 123°30'45.0" E),

– Olenyokskaya Channel, center: located on the first terrace in the SW part of the LRD (72°33'56.9" N, 125°03'52.3" E),

– Kurungnakh Island: located on the third terrace in an *alas* depression on Kurungnakh Island about 10 km SW of Samoylov Island (72°19'12.5" N, 126°11'35.7" E). The installation of the borehole destroyed the surface vegetation and thereby triggered melting of excess ground ice and the formation of a thermokarst pond around the borehole within one year (see Supplementary Material). The ground temperature record must therefore be considered disturbed and most

likely features a warm-bias compared to the surrounding undisturbed terrain. We therefore only employ the first three months of data following the drilling of the borehole.

– Sardakh Island: located in the SE part of the LRD near the main channel of the Lena River (72°19'12.6" N, 127°14'29.4" E). Sardakh is generally classified as part of the third terrace due to similar surface cover and height above river level, but the ground is actually comprised of neogene sandstone with a cover of Yedoma deposits (Kryamyarya et al., 2011).

At the borehole site, melting of excess ground ice has occurred since the installation of the borehole like in the case of Kurungnakh, which has led to subsidence of the surface and the formation of a pond around the borehole. This was observed for the first time in summer 2012 (see Supplementary Material) and we therefore exclude the later parts of the borehole record from the comparison to model results.

For the second terrace, there are no measurements of ground temperatures available.

Systematic measurements of thaw depths according to the CALM protocol have not been conducted outside Samoylov Island. However, there exist observations of thaw depths for single points in time and space for all three river terraces, facilitating

validation of regional differences in thaw depths:

- First terrace: In addition to the comprehensive record on Samoylov Island, a single measurement near the borehole site "Olenyokskaya Channel, center" is available from the year 2010.

- Second terrace: In summer 2005, thaw depths were recorded at several sites on Turakh Island (72° 56'24.4" N, 123°47'54.9" E) in the southwestern LRD near exposures at the shoreline and at a drill core site (Schirrmeister, 2007; Ulrich et al., 2009). Another manual thaw depth measurement was performed in the northern part of Arga Island (73°29'39.2" N, 124°22'33.1" E) in 2010. These observations are the only available ground truth information for the second terrace in the model period 2000-2014. Two additional observations are available from summer 1998 from the central part of Arga Island (73°20'18.5" N, 124°12'30.5" E) near Lake Nikolay and on Dzhipperies Island (72°51'14" N, 125°50'22" E) near Lake Yugus-Jie-Kuyele (Rachold and Grigoriev, 1999). While these cannot be compared to model output in a strict sense, they confirm the general order of magnitude of thaw depths on the second terrace.

- Third terrace: Thaw depth measurements are available from two distinct areas. At the W edge of the LRD, the thaw depth was recorded near the borehole site "Olenyokskaya Channel, mouth" in summer 2010. At three dates in July and August 2013, thaw depths were recorded at nine locations in the S part of Kurungnakh Island, near so-called "Lucky Lake" (72°17'41.0"N 126°9'34.0" E). The nine locations are contained within six 1 km model grid cells.

## 3 Methods

In this study, we update and extend the satellite data-based transient modeling of the ground thermal regime as outlined in Langer et al. (2013) to an area of approx. $16\,000\,\mathrm{km}^2$ within the LRD. The general idea is to employ time series of remotely sensed surface temperatures and snow depths to force a transient ground thermal model.

### 3.1 The CryoGrid 2 ground thermal model

CryoGrid 2 is a transient 1D ground thermal model based on Fourier's Law of heat conduction (Westermann et al., 2013). The model does not account for changing subsurface water contents due to infiltration and evapotranspiration, but instead assigns fixed values for the porosity and saturation of each grid cell. Freezing/thawing of soil water/ice is accounted for by a temperature-dependent apparent heat capacity (e.g. Jury and Horton, 2004) which is determined by the soil freezing characteristic according to the formulation by Dall'Amico et al. (2011). The apparent heat capacity and thermal conductivity of each layer are computed according to the volumetric fractions of water/ice (determined by the temperature), air and sediment matrix material composed of a mineral and an organic component. A more detailed description of the model physics and the numerical solvers is provided in Westermann et al. (2013).

CryoGrid 2 is capable of representing the annual build-up and disappearance of the snow cover by adding/subtracting grid cells according to a time series of snow water equivalent (which must be provided as part of the forcing data), but only allows for

constant thermal properties of the snow grid cells (both throughout the snow pack and over time). For this study, we assign a functional dependency between snow thermal conductivity $k_{\mathrm{snow}}$ and density $\rho_{\mathrm{snow}}$ according to Yen (1981):

$$k_{\mathrm{snow}} = k_{\mathrm{ice}} \left( \frac{\rho_{\mathrm{snow}}}{\rho_{\mathrm{water}}} \right)^{1.88}, \tag{1}$$

with $k_{\mathrm{ice}}$ and $\rho_{\mathrm{water}}$ denoting the thermal conductivity of ice and the density of water, respectively. This parameterization performed well over a wide range of snow densities and types in a dedicated validation study (Calonne et al., 2011). Furthermore, the snow density is employed to compute the volumetric heat capacity of the snow and to convert snow water equivalent to snow depth. As a result, the thermal properties of the snow pack are described by only a single parameter, the snow density $\rho_{\mathrm{snow}}$, for which an extensive set of in-situ observations is available from Samoylov island (Boike et al., 2013).

### 3.2 Subsurface properties and additional model parameters

At 1 km resolution, it is not possible to resolve small-scale differences of surface and subsurface properties. Therefore, we only distinguish the three river terraces as the main geomorphological units within the LRD for which we define "typical" subsurface stratigraphies based on available field observations (Sect. 2.1). The stratigraphies are provided in Table 1, while the boundaries of the terraces (Fig. 1) are based on Morgenstern et al. (2011) gridded to 1 km. For all terraces, a saturated bottom layer with mineral content of 70 vol.% is assumed, corresponding to densified fluvial deposits underlying the modern delta (Schirrmeister et al., 2011; Schwamborn et al., 2002b).

For the first terrace, a 0.15 m thick upper layer with high porosity and organic content is assigned, which is not entirely saturated with water or ice (Schneider et al., 2009; Langer et al., 2013). Below, the ground is assumed to be saturated, but the porosity remains high, corresponding to the ice-rich sediments. Based on field observations on Samoylov Island (Kutzbach et al., 2004; Zubrzycki et al., 2012), fine-grained silty sediments dominate the matrix material, with organic contents of approx. 5 vol. %. The depth of this layer is set to 9 m, based on observations for the depth of ice wedges in the first terrace (Schwamborn et al., 2002b). Note that these ground properties are also assigned to the active floodplain areas within the first terrace (Sect. 2.1), which cannot be meaningfully delineated at 1 km scale. In such floodplain areas, the model results must therefore be considered with care. Furthermore, the polygonal tundra landscape features a strong variability in surface soil moisture and vegetation/sediment conditions over distances of a few meters (Boike et al., 2013), which cannot be captured by the single stratigraphy employed for the modeling.

The sandy sediments of the second terrace largely lack an organic upper horizon (Rachold and Grigoriev, 1999; Ulrich et al., 2009; Schneider et al., 2009), so that a uniform upper layer with typical porosity of sand is prescribed (Table 1).

The third terrace is dominated by a relatively dry organic top layer with high porosity (Schneider et al., 2009; Zubrzycki et al., 2012), followed by a thick layer with very high ice contents (and organic contents of 5 vol. %), corresponding to the late Pleistocene Yedoma deposits (Schwamborn et al., 2002b; Schirrmeister et al., 2011). While the mineral fraction of this layer in reality is composed of fine-grained silty sediments, we assign "sand" as sediment type (Table 1) to account for the freezing characteristic of the extremely ice-rich ground which can be expected to resemble that of free water/ice rather than that of saturated silt.

The thermal conductivity of the mineral fraction of the sediment matrix required for the calculation of the soil thermal conductivity (Westermann et al., 2013) is set to $3.0\,\mathrm{W\,m^{-1}K^{-1}}$, as in previous modeling studies on Samoylov Island (Langer et al., 2011a, b, 2013). The sensitivity study by Langer et al. (2013) showed that the snow thermal properties are the most important model parameter controlling the simulated ground thermal regime. Therefore, the snow density (which controls both snow depth, heat capacity and thermal conductivity, Sect. 3.1) is a crucial parameter for which spatially or temporally distributed data sets covering the entire LRD are not available. However, an extensive set of measurements from polygonal tundra on Samoylov Island suggests snow densities of $(225\pm25)\,\mathrm{kg\,m^{-3}}$ (Fig. 6b, Boike et al., 2013) for polygon centers with well-developed snow cover, so that it is possible to explicitly account for the uncertainty of this important parameter by conducting model runs for a range of snow densities. For comparison to in-situ data (Sects. 4.1.1, 4.2.1), we present model runs with confining values of 200 and $250\,\mathrm{kg\,m^{-3}}$ (thus providing a range of ground temperatures), while the spatially distributed model runs (Sect. 4.2.2) are conducted with an average snow density of $225\,\mathrm{kg\,m^{-3}}$. Note that the confining values represent one standard deviation and that higher and lower snow densities occur regularly (Boike et al., 2013).

## 3.3 Model forcing data

CryoGrid 2 requires time series of surface (i.e. skin) temperatures and snow water equivalent as forcing data sets.

*Surface temperature:* As temperature forcing at the upper model boundary, a product synthesized from clear-sky land surface temperatures (LST) from the "Moderate Resolution Imaging Spectroradiometer" (MODIS) and $2\,\mathrm{m}$ air temperatures from the ERA–interim reanalysis (Dee et al., 2011) was applied. For this purpose, the daily MODIS level 3 LST products MOD11A1/ MYD11A1 in the version 005 were employed, which deliver four LST values per day (Terra and Aqua satellites, day and night time LST each). The merging procedure is similar as described in Westermann et al. (2015) in which spatially distributed data sets of freezing and thawing degree days were generated. In essence, gaps in the MODIS LST record due to cloud cover are filled by the the reanalysis data, which creates a data record with homogeneous data density and has the potential to moderate the cold-bias of temporal averages of surface temperatures computed from clear-sky MODIS LST (Westermann et al., 2012, 2015). During cloudy skies, differences between air and surface temperatures are strongly reduced compared to clear-sky conditions (e.g. Gallo et al., 2011), so that air temperatures can be regarded an adequate proxy when MODIS LST is not available due to cloud cover. Note that this gap-filling procedure assumes that air temperatures from the ERA reanalysis are not strongly biased. For melting snow, surface temperatures are confined to the melting point of ice, while air temperatures can be positive. Positive values of the surface temperature forcing are therefore set to $0°\mathrm{C}$ if a snow cover is present (see below). For this study, we create a time series of weekly averages of surface temperatures to force the CryoGrid 2 model. The reanalysis data, which are available at $0.75°$ resolution, are interpolated to the center point of each MODIS LST pixel (in the sinusoidal projection native to MOD11A1/MYD11A1 data). The satellites carrying the MODIS instrument were launched in 2000 (Terra) and 2002 (Aqua), respectively, while ERA–interim reanalysis is available since 1979. The synthesized time series used for model forcing therefore extends from 15 May 2000 to 31 October 2014 and thus covers the period for which remotely sensed LST data from at least one satellite are available. For the first two years, the data density of MODIS LST measurements in the composite product is lower than after summer 2002 when LST measurements from Aqua become available. Spatially, the fraction of the

successful MODIS LST retrievals is relatively constant throughout the LRD, varying between 50 and 55%. In summer and fall, retrieval fractions are generally lower (40-50%) than winter and spring (55-70%), indicating more frequent cloudy conditions in summer and fall.

*Snow depth:* Similar to the procedure outlined in Langer et al. (2013), a weekly snow water equivalent (SWE) product was synthesized from GlobSnow SWE (Pulliainen, 2006) (25 km resolution) and the MODIS level 3 Snow Extent (SE) products MOD10A1/MYD10A1 (0.5 km resolution), which for clear-sky conditions deliver two values of binary flags (1: snow; 0: no snow) per day (one for Terra and Aqua each). The latter products were averaged over the 1 km sinusoidal grid of the MODIS LST data and the two satellites, yielding a number between 0 and 1 for each day with available data, corresponding to the fraction of successful retrievals at the 0.5 km pixel level flagged as "snow". We then applied a "maximum change" detection algorithm to the data set to determine the most likely dates for the start and the end of the snow cover in each 1 km pixel. For this purpose, we compute the fractions of 1 km values with values of 0 and 1, respectively, both within a window of four weeks before and after each date. The snow start date is determined as the date for which the sum of fractions of 0 before and fractions of 1 after is largest. This sum can be up to 2 when there are 100% retrievals flagged as snow-free before and 100% retrievals flagged snow-covered before the date. For the snow end date, the opposite criterion is applied, i.e. the sum of the fractions of 1 before and fractions of 0 after features a maximum. Note that the large window is required as prolonged cloudy periods often occur in the study area, for which no measurements are available. The MODIS SE products cover the same periods as the MODIS LST data (see above).

GlobSnow SWE (Daily L3A SWE, level 2.0) data are derived from passive microwave remote sensors, which are not affected by clouds, so that a gap-free daily time series is in principle available for entire model period from 2000 to 2014. The GlobSnow processing algorithm is based on a data assimilation procedure, which also takes in-situ measurements at WMO (World Meteorological Organization) stations into account (Takala et al., 2011). For the LRD, the closest station is located at Tiksi, about 50 km to the E, while the closest stations to the W are several hundred kilometers away. The station measurements are interpolated in space to obtain a SWE background field which is then weighted against SWE information derived from the passive microwave sensor by means of forward modeling of snowpack microwave emission using the HUT model (Pulliainen et al., 1999). In the data assimilation procedure, a spatially constant snow density of 240 kg m$^{-3}$ is assumed, which is in the range of the in-situ measurements on Samoylov island (Sect. 3.2).

The SWE values in the LRD (see Sect. 4.1) are typically below the critical threshold of about 150 mm above which SWE can no longer reliably derived from passive microwave retrievals (Takala et al., 2011). On the other hand, SWE retrieval is hampered for shallow snow cover and for wet melting snow, so that the start and the end of the snow season is not well covered by GlobSnow. Furthermore, water bodies constitute a major error source (e.g. Derksen et al., 2012) and generally lead to underestimation of SWE, in particular when the ice cover is thin (Lemmetyinen et al., 2011). Due to admixing of microwave radiation emitted from the ocean, the number of SWE retrievals is very small or even zero in the coastal areas of the LRD, so that almost half of the area of the LRD could not be included in the modeling. The boundary of the final model domain was finally chosen so that all validation sites (Fig. 1) are located within. In a few cases (in particular the sites AN, Tu and OM, Fig. 1), the available SWE data had to be extrapolated by about one grid cell or 25 km, which seems adequate considering the

smoothness of the remote sensing derived SWE field in the LRD.

As a first step, the daily SWE data were interpolated from the Northern Hemispherical EASE-Grid projection (25 km resolution) to the 1 km sinusoidal grid of the MODIS LST data. We subsequently assign linearly increasing SWE from the date identified as the most likely snow start date (using the MODIS SE product, see above) and the next available GlobSnow SWE measurement. The same procedure is applied for the snow end date. Note that this procedure can result in a step-like increase or decrease of the snow depth, if a valid GlobSnow SWE value is available for the identified start/end date. As a final step, the daily time series is averaged to the same weekly periods as the employed surface temperature forcing (see above) and SWE converted to snow depth with the applied snow density (Sect. 3.2). The use of medium-resolution MODIS SE facilitates correcting the coarse-scale GlobSnow SWE product regarding the start and the end of snow cover period, both of which can crucially influence the modeled ground thermal regime. Nevertheless, passive microwave-derived SWE is associated with considerable uncertainty in the LRD. We therefore compare the model snow forcing to in-situ measurements from Samoylov Island (Sect. 4.1.1) and to independent spatial SWE data sets (Sect. 4.1.2, Supplementary Material).

## 3.4 Model set-up

For each 1 km grid cell, the ground thermal regime was simulated for a specific ground stratigraphy and forcing time series of surface temperatures and snow depths. In the vertical direction, the ground between the surface and 100 m depth is discretized in 163 layers, which increase in size from 0.02 m near the surface (until 1.5 m depth so that the active layer is modeled at maximum resolution) to 10 m near the bottom, similar to the set-up in Westermann et al. (2013). Within the snow cover, the minimum layer size of 0.02 m is prescribed. At the lower boundary, a constant geothermal heat flux of $50\,\mathrm{mWm^{-2}}$ is assumed, as estimated from a 600 m deep borehole 140 km east of Samoylov Island (Langer et al., 2013).

To estimate a realistic initial temperature profile, a model spin-up is performed to achieve steady-state conditions for the forcing of the first five model years, using the multi-step procedure outlined in detail in Westermann et al. (2013). In a first step, the model is run to estimate the average temperature at the ground surface (i.e. below the snow cover in winter), for which the steady-state temperature profile in the ground is assigned to all grid cells (considering the geothermal heat flux at the bottom and the thermal conductivity of all grid cells). In a second step, CryoGrid 2 is run twice for the first five model years, so that the annual temperature cycle to the depth of zero annual amplitude is reproduced. The simulations for the entire time series can thus be initialized by a temperature profile that is both adequate for the upper and the lower parts of the model domain. We emphasize that the initialization procedure limits the CryoGrid 2 results to the uppermost few meters of the soil domain since deeper temperatures are still influenced by the surface forcing prior to the model period, for which satellite measurements and thus model forcing data are not available.

# 4 Results

## 4.1 Forcing data sets

### 4.1.1 Comparison to in-situ data

Systematic in-situ observations on surface temperature and snow depths are only available for the Samoylov permafrost observatory, so that a validation of the spatial patterns of the model forcing data within the LRD is not possible.

*Surface temperature*: We compare the surface temperature forcing synthesized from MODIS LST and ERA reanalysis air temperatures (Sect. 3.3) to measurements of surface (skin) temperature from Samoylov Island from 2002 to 2009 (Boike et al., 2013). The results of the comparison for the 1 km grid cell in which the observation site is located, are displayed in Fig. 2: while the annual temperature regime is reproduced very well, a systematic cold-bias of on average -0.8°C remains which is consistent throughout the year. Fig. 2 (bottom) also shows a comparison of monthly averages of all available MODIS LST measurements, i.e. without filling the gaps in the time series with ERA reanalysis air temperatures. Here, a significantly larger cold-bias of up to 3°C is found for all months except July, which is in line with validation studies from Svalbard which demonstrate a similar cold-bias during the winter moths (Westermann et al., 2012; Østby et al., 2014). In July, the average of all MODIS LST measurements is significantly warmer than the observations. However, surface temperatures can feature a strong spatial variability during summer due to differences in surface cover and soil moisture conditions (Langer et al., 2010; Westermann et al., 2011b), so that the scale mismatch between the 1 km remotely sensed LST values and the in-situ point observations may explain at least part of the deviation. In summary, the time series of surface temperatures synthesized from MODIS LST and ERA-interim reanalysis air temperatures facilitates an adequate representation of in-situ observations and thus well suited as input for ground thermal modeling (at least in homogeneous terrain), which supports earlier results from the N Atlantic permafrost region (Westermann et al., 2015). However, the slight, but systematic cold-bias must be taken into account when analyzing the uncertainty of modeled ground temperatures.

*Snow cover*: As for surface temperatures, only point measurements on Samoylov Island are available for snow depth which are compared to the forcing time series of snow water equivalents synthesized from 25 km GlobSnow SWE and 0.5 km MODIS SE (Sect. 3.3). In general, snow depths computed from GlobSnow SWE with snow densities between 200 and 250 kg m$^{-3}$ can reproduce the order of magnitude of the in-situ measurements, with differences generally smaller than 0.1 m (Fig. 3). At least some of the observed interannual differences are reproduced in the remote sensing-derived snow product, e.g. the above-average snow depths in winter 2003/04 and the below-average snow depths in 2012/13 (the latter was qualitatively noted by the station personnel, pers. comm., N. Bornemann). For values with non-zero snow depth, the model forcing (using a snow density of 225 kg m$^{-3}$) features an RMSE of about 0.06 m, and a slight positive bias of 0.015 m. The average snow depth in polygonal tundra (obtained by a spatially distributed survey, Boike et al., 2013) in early spring 2008 is slightly higher than both point measurements from the snow depth sensor and the model forcing. However, the difference is only about 0.05 m for the model forcing with snow density 225 kg m$^{-3}$, well within the observed spatial variability of snow depths (Fig. 3).

Start and end dates of the snow cover are compared to in-situ observations (Fig. 4) based on interpretation of time-lapse im-

agery from an automatic camera system (Boike et al., 2013). The snow melt date, which is crucial for capturing the onset of soil thawing correctly, is generally well captured, although differences of more than half a month exist for some of the years. We emphasize that the transition from a completely snow covered to a completely snow-free surface occurs over an extended period of time due to spatially variable snow depths, so that a "snow melt date" in a strict sense does not exist. The MODIS SE processing algorithm based on surface reflectances may apply a different threshold for the characterization of a snow-free surface than the subjective interpretation of the in-situ camera images. Furthermore, prolonged periods of cloudiness make

remote detection of snow cover impossible, so that a considerably reduced accuracy must be expected in such years. The same issues apply to the detection of the snow start date. While deviations of more than 15 days exist in the beginning of the period, the remotely detected snow start date in general follows the in-situ observations (Fig. 4). We conclude that the model forcing can reproduce the general magnitude of snow depths on Samoylov Island, as well as the timing of the snow-covered season, at least for the majority of the considered years. However, due to the considerable uncertainties associated with GlobSnow

SWE retrievals (Takala et al., 2011) the snow depth model forcing for the entire LRD must be considered less reliable than the surface temperature forcing.

### 4.1.2   Spatial distribution in the LRD

Fig. 5 displays the spatial distribution of yearly average surface temperatures (b), freezing degree days (c), thawing degree days (d), snow-free days (e) and average snow depth (f) for a ten-year period 2004-2013, as well as the classification of subsurface

stratigraphies (a, see Sect. 3.2). Average surface temperatures feature only moderate spatial differences in the order of $2°C$, with the warmest areas close to the main river channels in the southern part of the LRD. Similarly, the differences in freezing degree days are only on the order of 10 to 15%, with the largest number of freezing degree days recorded in the central parts of the LRD, which is located furthest away from the coastline and main river channels. On the other hand, thawing degree days feature a pronounced north-south gradient, with values almost twice as large in the southern parts of the LRD compared to the

areas at the north coast. A similar pattern is found for the average number of snow-free days which varies between around 100 in the northern areas and around 140 in the southern areas.

Average snow depths are largest in the western areas and decrease towards the southeastern parts of the LRD, although the differences are only small. This spatial distribution is in coarse agreement with Canadian Meteorological Centre (CMC) Snow Depth Analysis Data (Brasnett, 1999), an independent global snow product at $24 \, \text{km}$ resolution based on precipitation data from

an atmospheric model (see Supplementary Material). As passive microwave data are not employed in the CMC Snow Depth Reanalysis, the match is an indication that the overall snow depth pattern in Fig. 5f is not an artifact of the GlobSnow retrieval algorithm, but rather reflects spatial differences in snowfall. This conclusion is further supported by winter precipitation from the ERA-interim reanalysis which also displays a west-east gradient over the land areas in the LRD (see Supplementary Material). However, we emphasize that the effective spatial resolution of the remotely sensed snow depth data is significantly

coarser than for the other variables, so that large biases are likely to occur at the model scale of $1 \, \text{km}$, at least for single grid cells. Furthermore, the quality of the SWE retrievals is insufficient in coastal areas (Sect. 3.3) which hence are not covered by the ground thermal modeling.

## 4.2 Modeled ground thermal regime

### 4.2.1 Comparison to in-situ data

The model results are validated for ground temperatures and thaw depth for nine field sites, Samoylov Island, Olenyokskaya Channel center and mouth, Arga Island north and center, Dzhipperies Island, Turakh Island, Kurungnakh Island and Sardakh Island (Fig. 1, Sect. 2.2). With this data basis, all three stratigraphic classes are covered by two or more in-situ measurement sites. However, for the second terrace only few unsystematic thaw depth measurements are available and observations of ground temperatures are lacking entirely.

*Ground temperature*: To assess modeled ground temperatures, we use in-situ measurements of active layer temperatures from Samoylov Island (first terrace), as well as measurements of permafrost temperatures at 2-3 m depth in boreholes. At this depth, the temperature regime is dominated by the surface forcing over a couple of square meters surface area which averages over smaller-scale variability of surface and subsurface properties. On the other hand, the modeled temperature field is not strongly dominated by the initial condition, at least after the first years of simulation.

Fig. 6 displays a comparison of modeled and measured active layer temperatures at 0.4 m depth in a wet polygon center on Samoylov Island in the first terrace. In general, the in-situ values are contained within the range of modeled ground temperatures for the two confining snow depths, but some deviations exist during refreezing in fall. In a few years, the length of the so-called "zero-curtain" when temperatures remain in the vicinity of $0°C$ is underestimated in the simulations. Possible reasons are a too high thermal conductivity of the uppermost, already frozen soil layers, higher than average surface temperatures in the more moist sites during refreezing (compare Langer et al., 2010), or a shallow snow or rime cover at the surface which is not detected by remote sensors.

Although small, a similar effect is visible in several years for the modeled temperatures in shallow boreholes on the first and third terrace (Fig. 7) for which the pronounced cooling in fall occurs too early in the model runs. The consistent occurrence at several locations in the LRD points to a shortcoming of the model scheme rather than local conditions, e.g. caused by spatial variability of the subsurface properties. Despite such problems, the model scheme allows an adequate representation of measured ground temperatures within the range of uncertainty due to the snow density, except for the periods when thermokarst development around the boreholes was evident (shaded grey in Fig. 7). The 26 m deep borehole on Samoylov Island (Boike et al., 2013) is located near the south-west edge of the island in a relatively well-drained environment. With the relatively water- and ice-rich stratigraphy used for the first terrace (Table 1), considerably colder ground temperatures are modeled compared to the measurements (Fig. 8 left), particularly during summer and fall. Using the same surface forcing, but a stratigraphy oriented at the true conditions at the borehole (sandy sediments; 0-0.5 m: 30 vol. % water/ice, 10 vol. % air, 60 vol. % mineral; 0.5-9 m: 40 vol. % water/ice, 60 vol. % mineral; deeper layers as for first terrace) significantly improves the match between modeled and measured values, especially during summer (Fig. 8 right).

A comparison of monthly averages for all five boreholes is shown in Fig. 9. For a snow density of $225\,\mathrm{kg\,m^{-3}}$, the model results feature an RMSE of $1.1°C$ and an average bias of $-0.9°C$, mainly due to underestimation of measured values during the summer and fall seasons. For a snow density of $200\,\mathrm{kg\,m^{-3}}$, the model bias is on average positive ($+0.8°C$), but the RMSE is increased

(1.6°C). The model performance is worst for the highest snow density (RMSE 2.1°C, bias -2.1°C). If the Samoylov Island borehole (for which the ground stratigraphy was adjusted, see above) is removed, the model performance for the best-fitting snow density of $225 \, \text{kg m}^{-3}$ remains largely unchanged (RMSE 1.2°C, bias -0.9°C). Fig. 10 displays an inter-site comparison of measured and modeled yearly average ground temperatures for a two-year period for which largely gap-free in-situ records from four sites are available. All measurements are contained in the range of modeled ground temperatures for the confining snow densities of 200 and $250 \, \text{kg m}^{-3}$, although the in-situ value for Sardakh is located near the upper bound of the modeled

temperature range. For the average snow density of $225 \, \text{kg m}^{-3}$, the measured and modeled values agree within 1 to 1.5°C, which can serve as a coarse accuracy estimate for the spatially distributed simulations of the ground thermal regime in the LRD (Fig. 12, see Sect. 4.2.2). If snow densities are allowed to vary between 200 and $250 \, \text{kg m}^{-3}$, the agreement is generally better than 2°C. While the model performance is encouraging, we emphasize that it is mainly based on only four sites (the Kurungnakh record comprises only a short period) which are all located in the southern part of the LRD.

*Thaw depth*: In the LRD, temporally resolved measurements of thaw depths are only available from Samoylov Island. Fig. 11 compares modeled thaw depths with the average of 150 points for which thaw depths have been measured manually over a period of 13 years (Boike et al., 2013). In general, the model scheme can represent the measured thaw depths very well, with deviations of 0.1 m or less. In particular in the second half of the model period, the agreement is excellent with deviations of 0.05 m or less. Furthermore, the annual dynamics of the thaw progression are adequately resolved. We emphasize that the

in-situ measurements are evidence of a considerable spatial variability of thaw depths even, with an average standard deviation of 0.06 m. This variability is not captured by the model runs with different snow densities, which only induces differences in modeled thaw depths of a few centimeters Fig. 11. These results are in agreement with the sensitivity analysis of Langer et al. (2013) who showed for Samoylov Island that ground temperatures are most sensitive to snow thermal properties, while the thaw depth is more dependent on ground properties and ice contents, which are set constant in the simulations (Table 1).

The comparison of modeled and measured thaw depths for the point measurements in the three stratigraphic units of the LRD is shown is Table 2. The in-situ observations are clear evidence that thaw depths are by far shallowest for the third terrace, while the largest thaw depths occur in the second terrace. The model scheme can reproduce this pattern very well, although deviations between measured and modeled thaw depths of 0.1 m or more can occur. The largest deviations occur for Turakh Island for which the model significantly underestimates the measured thaw depths. However, the measurements were performed near

terrain edges and at slopes (Schirrmeister, 2007), so that a reduced match must be expected when comparing to thaw depths obtained for the simplified "model case" of flat homogeneous terrain. All in all, the comparison suggests that the presented model scheme accounts for the main drivers of active layer dynamics and can reproduce systematic differences in thaw depths between the main geomorphological units in the LRD.

### 4.2.2   Spatial distribution in the LRD

Fig. 12 presents average ground temperatures at 1.0 m depth (i.e. well below the active layer, see next section) for the ten-year period 2004-2013. Within each stratigraphic unit, modeled ground temperatures generally decrease from west to east, following the spatial pattern of snow depth in the LRD (Fig. 5), and towards the North, presumably as a result of low summer surface

temperatures and shorter snow-free period (Fig. 5). At the same time, the ground stratigraphic units have a pronounced impact on modeled ground temperatures, with lowest temperatures modeled for the third and warmest for the second terrace (compare Fig. 12). This is corroborated by the results of a sensitivity analysis towards the ground stratigraphy for the nine validation sites in the LRD (Table 3). When using the same forcing data, but different ground stratigraphies, the modeled ground temperatures are generally lowest for the third terrace and highest for the second terrace stratigraphy.

The highest ground temperatures are modeled for parts of the second terrace in the northwest and for the areas around the
Olenyokskaya Channel in the southwest part of the LRD where ground temperatures higher than -9°C are mapped. Medium temperatures of -9 to -11°C are obtained for the center of the delta and thus large parts of the first terrace. In the eastern part of the LRD, the lowest average temperatures with less than -11°C are modeled for parts of the third terrace.

*Thaw depth*: The spatial distribution of modeled maximum thaw depths (Fig. 13) is mainly related to two factors: the thawing degree days, which decrease strongly from south to north (Fig. 5) in the LRD, and the ground stratigraphy. For the third
terrace, average maximum thaw depths of less than 0.3 m are modeled, while the second terrace features maximum thaw depths of 0.65 to 0.95 m. In the first terrace, the modeled thaw depths are largest in the southern part (approx. 0.5 m), while the northeastern part feature considerably lower maximum thaw depths that are of similar magnitude as for the third terrace (0.3 m). These results are in agreement with the sensitivity analysis for the validation sites (Table 3), which clearly shows the strong dependence of modeled thaw depths on the ground stratigraphy.

## 5   Discussion and Outlook

### 5.1   Model forcing

#### 5.1.1   Surface temperature

Validation studies have revealed a significant cold-bias of long-term averages derived from MODIS LST in Arctic regions (Westermann et al., 2012; Østby et al., 2014), which is attributed to the over-representation of clear-sky situations and defi-
ciencies in the cloud detection during polar night conditions (Liu et al., 2004). The same bias is found for Samoylov Island (Fig. 2) for which averages directly computed from MODIS LST measurements are cold-biased by about 1-2°C for most of the year. In this study, we therefore employ a gap-filling procedure with ERA-interim near-surface air temperatures. During cloudy periods, reanalysis-derived air temperatures may indeed facilitate an adequate representation of surface temperatures, as the near-surface temperature gradient is smaller compared to clear-sky conditions (e.g. Hudson and Brandt, 2005; Gallo et al.,
2011; Westermann et al., 2012).

As demonstrated by Westermann et al. (2015) for the N Atlantic region, the composite product features a considerably reduced bias and is significantly better suited as input for permafrost modeling than the original MODIS LST record. However, a small, but consistent cold-bias of about 0.8°C remains. This could be explained by the fact that the gap-filling procedure only applies to gaps due to clouds that are successfully detected, but does not remove strongly cold-biased LST measurements of cloud
top temperatures (Langer et al., 2010; Westermann et al., 2011b) that regularly occur when the MODIS cloud detection fails.

Here, further improvements seem feasible, e.g. through simple plausibility criteria when comparing the remotely sensed LST against meteorological variables of the ERA-reanalysis data set. However, such methods are most likely sensitive towards a range of factors, such as landcover and exposition (which strongly influence the true surface temperature), so that they should be carefully developed and validated for a range of sites. Based on in-situ measurements, Raleigh et al. (2013) suggest that for snow-covered ground dew point temperatures are a better approximation for surface temperatures compared to air temperatures at standard height. However, observations on Samoylov Island display only a small offset between snow surface and air temperatures, with the difference increasing from near zero in early winter to about $1°C$ in late winter (Table 3, Langer et al., 2011b). The reason for this is most likely that the ground heat flux is a strong heat source especially in early winter (Langer et al., 2011b) which warms the surface and thus prevents formation of a strong near-surface inversion. Therefore, we consider air temperatures an adequate proxy for snow surface temperatures in the LRD, but dew point temperatures should clearly be considered for gap-filling in the snow-covered season in future studies. We conclude that surface temperatures synthesized from MODIS LST and near-surface air temperatures from the ERA-interim reanalysis are an adequate choice for the purpose of ground thermal modeling in the LRD, at least in homogeneous terrain, although it may introduce a slight cold-bias in modeled ground temperatures.

### 5.1.2 Snow

As demonstrated by Langer et al. (2013), snow depth and snow thermal properties are crucial factors for correctly modeling ground temperatures in the LRD. In this light, the coarsely resolved estimates of GlobSnow SWE must be considered the key source of uncertainty for the thermal modeling.

– The performance of GlobSnow SWE has been evaluated on continental scales by comparison to systematic in-situ data sets (Luojus et al., 2010; Takala et al., 2011). For Eurasia, surveys spanning the entire snow season (Kitaev et al., 2002) were compared from 1979 to 2000. For shallow snow (approx. SWE<60 mm), GlobSnow SWE tends to overestimate observed values slightly, but the relationship between measurements and GlobSnow retrievals is on average linear. When SWE exceeds approx. 100 mm, the GlobSnow algorithm tends to underestimate measured SWE, and for values larger than 150 mm the signal from passive microwave retrievals saturates and SWE can no longer reliably be detected (Takala et al., 2011). For the LRD, both in-situ measurements and GlobSnow values indicate that SWE is generally below this critical threshold so that saturation effects most likely do not play a role for the uncertainty. The Eurasia data set is strongly biased towards sites in steppe environments and the boreal forest zone (where SWE retrieval is affected by the canopy, e.g. Derksen et al., 2012), while northern tundra areas with characteristics similar to the LRD are strongly undersampled. A more representative data set is available from an extensive transect across Northern Canada (Derksen et al., 2009), for which comparison of GlobSnow SWE retrievals yielded an RMSE of 47 mm and an average bias of -36 mm. The average SWE of 120 mm (Takala et al., 2011) was significantly larger than in the LRD, so that it is not meaningful to transfer the absolute uncertainties. When using relative uncertainties, on the other hand, we arrive

at a similar RMSE as for the comparison of the time series on Samoylov Island (0.06 m, see Sect. 4.1.1): for N Canada, a relative RMSE of around 40% was found, which corresponds to an absolute RMSE of 0.065 m in snow depth, when scaled to the average of around 0.16 m on Samoylov Island (Fig. 5f). Although the character of the two data sets differs (spatial transect vs. multi-year point measurement), the good agreement is an indication that the GlobSnow performance in the LRD could be similar to N Canada. We emphasize that the RMSE corresponds to undirected fluctuations around the average value which have much less influence on the modeled average ground thermal regime (Figs. 12, 13) than a systematic bias.

– Water bodies strongly affect microwave emission of the ground, which is known to lead to underestimation of SWE in passive microwave-based retrievals (Rees et al., 2006; Lemmetyinen et al., 2011). For the above mentioned N Canada data set, water bodies might explain the significant bias of 36 mm (Takala et al., 2011), but the average values (120 mm) are also sufficiently high that saturation effects (Luojus et al., 2010) are likely to contribute to the bias. In the LRD, water bodies are abundant features (Fig. 1), so that GlobSnow retrievals are likely to be affected. Using a Landsat (Schneider et al., 2009) and MODIS (MODIS water mask) based land cover classifications, we estimate the water fraction in the employed 25 km grid cells in the Lena River Delta to be between 12 and 30%, with a single grid cell in the E part reaching 37% (of which more than half is estimated to be river arms, see below). Almost three quarters of the grid cells feature water fractions of less than 20%. However, relatively shallow themokarst lakes dominate in the LRD, which at least partly freeze to the bottom in winter (Schwamborn et al., 2002a; Antonova et al., 2016), so that microwave emission becomes similar to land areas, although in particular the wave-length dependency of the effect may be complex (Gunn et al., 2011). Furthermore, the winter discharge of the Lena River is very low compared to other northern rivers, as the catchment is largely located in the continuous permafrost zone (Yang et al., 2002). We estimate the winter discharge to be only about 10% of summer averages (Fig. 2 in  Yang et al., 2002), and large river areas identified as water in summer-derived satellite imagery must fall dry in winter, which decreases the water fraction in the central and eastern part of the delta (where the water fractions are highest) considerably. Furthermore, also shallow river arms and even coast-near areas of the Laptev Sea (Eicken et al., 2005) freeze to the bottom, so that we expect the true "open water" fraction relevant for microwave emission in winter to be significantly lower than the open water fractions obtained from summer imagery (see above) suggest. This is corroborated by the comparision to in-situ measurements for Samoylov Island (Fig. 3) situated in a relatively water-body-rich area where we find a satisfactory performance for GlobSnow. The largest impact on SWE retrievals is most likely during lake freezing and snow cover build-up in fall, when GlobSnow SWE retrievals must be considered highly uncertain. In the future, enhanced SWE retrieval algorithms taking the effect of water bodies explicity into account (e.g. Lemmetyinen et al., 2011) may become available.

– The spatial resolution of 25 km is insufficient to capture the considerable spatial variability of snow depths in the LRD, both on the modeling scale of 1 km and the considerably smaller scales where the snow distribution is strongly influenced by the microtopography (Boike et al., 2013). Studies with equilibrium models have demonstrated that the latter can to a certain degree be captured by statistical approaches that employ an (estimated) distribution of snow depths to obtain

distributions of ground temperatures for each grid cell (Gisnås et al., 2014, 2015; Westermann et al., 2015). However, with the transient modeling scheme employed in these study, new issues arise that strongly complicate the application of a statistical representation of snow cover. First, spatial differences in snow depth will inevitably lead to a different timing of the snow melt which could influence in particular the modeled active layer thickness. Such small-scale differences of the snow start date cannot be captured by the 0.5 km scale MODIS SE product. Secondly, it is not clear how the distribution of snow depths can be translated to forcing time series of snow depths that are required for the CryoGrid 2 modeling. In some areas, snow depths may be relatively constant from year to year, while there may be strong interannual variations at other sites. Such temporal evolution is not contained in the distribution of snow depths, and computationally demanding deterministic snow redistribution models (e.g. Lehning et al., 2006) may be required to overcome such problems.

– In the coastal regions of the LRD, GlobSnow SWE does not provide a sufficient number of retrievals, so that the annual dynamics of the snow cover can be captured. In general, these regions must be excluded from the model domain. In this study, we chose to extrapolate the GlobSnow SWE retrievals to adjacent regions, so that more validation sites could be covered. The same issue applies to regions with pronounced topography which precludes the use of the modeling scheme for mountain permafrost area.

– The snow density is a crucial parameter, as it controls both the snow depth (since SWE is used as driving input data), the snow volumetric heat capacity and the snow thermal conductivity. In this study, the snow density was assumed to be constant in time and space, with the values determined by in-situ measurements (similar to Westermann et al., 2013; Langer et al., 2013). While this may be adequate for the relatively small model domain of the LRD, spatially distributed information on typical snow densities (e.g. Sturm et al., 1995) would be required for application on larger scales.

– The end and start of the snow cover have been determined at a comparatively high spatial resolution of 1 km using the MODIS SE product (Fig. 4), which corresponds to a downscaling of the coarsely resolved GlobSnow SWE product for these important periods. Furthermore, the performance of the GlobSnow SWE product is relatively poor for very shallow snow depths and for wet (melting) snow (Pulliainen, 2006) which is to a certain extent moderated by prescribing the snow start and end dates.

## 5.2 The CryoGrid 2 model

In this study, CryoGrid 2 is employed for a relatively short period of approx. 15 years, so that the model initialization deserves a critical discussion (Westermann et al., 2013). A model spin-up to periodic steady-state conditions was performed for the first five years of forcing data, i.e. from summer 2000 to summer 2005. Ground temperatures in deeper soil layers are strongly influenced by the choice of the initial condition, and the modeled temperatures should not be interpreted further. Therefore, we restrict the comparison to in-situ measurements to the uppermost three meters of soil and for the period following 2002 for active layer measurements (Figs. 6, 11) and after 2006 for ground temperatures in 2-3 m depth (Figs. 7, 8). In both cases, the model results are sufficiently independent of the initialization (Langer et al., 2013) which must therefore be considered a minor source of uncertainty.

The applied ground stratigraphy has a significant direct influence on the simulations results, both on ground temperatures and thaw depths (compare Westermann et al., 2016). For this study, three landscape units with associated "typical" stratigraphies were defined, which facilitate capturing the observed large-scale differences in particular for the thaw depth (Sect. 4.2.2). However, a significant small-scale variability of ground properties is superimposed on these large-scale differences giving rise to a significant variability of thaw depths and ground temperatures that are not captured at 1 km scale. An example is the in-situ record of thaw depths measurements at 150 points on Samoylov Island for which the model scheme can capture the interannual

variations of the mean very well (Fig. 11). However, with an average standard deviation of 0.06 m the measurements feature a considerable spread (Boike et al., 2013) that is most likely explained by small-scale differences in ground properties, surface temperature and possibly snow cover. Another example is the borehole site on Samoylov Island, for which the "typical" ground stratigraphy for the first terrace is clearly not applicable (Fig. 8). In principle, such subgrid effects could be captured by running the model scheme not only for a single realization per grid cell, but for an ensemble of model realizations reflecting

the statistical distribution of ground stratigraphies and properties within a grid cell. Such a scheme could also be extended to account for a subgrid distribution of snow depths by assigning different snow depths (according to a defined distribution, e.g. Gisnås et al., 2015) to the ensemble members. In addition to a considerable increase in computation time (e.g. a factor of 100 for 100 ensemble members), field data sets with statistical information on ground stratigraphies are generally lacking for the LRD. A simpler way could be aggregating high-resolution landcover data sets (e.g. Schneider et al., 2009) to the 1 km grid,

so that fractional information on the landcover can be obtained. Assuming that each landcover class can be assigned a typical subsurface stratigraphy, the model scheme could be run for all landcover classes/stratigraphies present within one 1 km grid cell.

The model physics of CryoGrid 2 does not account for a range of processes that may influence the ground thermal regime in permafrost areas, such as infiltration of water in the snow pack and soil (Weismüller et al., 2011; Westermann et al., 2011a;

Endrizzi et al., 2014), or thermokarst and ground subsidence due to excess ground ice melt. The latter can strongly modify the ground thermal regime, as demonstrated by Westermann et al. (2016), which makes a comparison of model results to in situ measurements at thermokarst-affected sites (Kurungnakh, Sardakh, Sect. 4.2.1) challenging. Furthermore, small water bodies and lakes can strongly modify the ground thermal regime both in the underlying ground and in the surrounding land areas (Boike et al., 2015; Langer et al., 2015), so that the model results are questionable in areas with a high fraction

of open-water areas (Muster et al., 2012). While more sophisticated model schemes (Plug and West, 2009; Westermann et al., 2016) can simulate the ground thermal regime of such features, a spatially distributed application is challenging: in general, higher-complexity models require additional input data and model parameter sets (e.g. precipitation for a water balance model, Endrizzi et al., 2014) for which the spatial and temporal distributions are poorly known. Furthermore, the model sensitivity may vary in space depending on the interplay of different model parameters and input data (Gubler et al., 2013) which makes

it harder to judge the uncertainty of model results.

## 5.3 The modeled ground thermal regime

The validation results suggest a model accuracy of 1 to 2°C for multi-annual average ground temperatures (Fig. 10) and around 0.1-0.2 m for annual maximum thaw depths (Table 2). On the one hand, high ground temperatures are modeled along the large river channels in the southern part of the LRD. These areas also feature high average surface temperatures (Fig. 5) which could at least partly be related to warm water advected by the Lena river. Surface temperatures derived from remote sensors have a significant advantage over data sets derived from atmospheric modeling, which in general cannot reproduce

such effects. On the other hand, the modeled ground temperatures are clearly influenced by ground stratigraphy. As evident in Fig. 12, the second terrace is systematically warmer than the adjacent first terrace, which is not visible in the temperature forcing (Fig. 5). This finding is corroborated by the sensitivity analysis (Table 3) which showcases the importance of a sound representation of ground thermal properties, in particular in and just below the active layer, for correct modeling of ground temperatures. These differences are at least partly related to stratigraphy-dependent thermal offsets between average ground

surface and ground temperatures caused by seasonal changes of subsurface thermal conductivities due to freezing and thawing (Osterkamp and Romanovsky, 1999).

Thaw depths are to an even larger extent determined by the ground stratigraphy. On the third terrace, a comparatively dry organic-rich layer with low thermal conductivity limits the heat flux so that the underlying ice-rich layers experience only a limited amount of thawing. As a consequence, the thaw progression hardly extends below the uppermost layer, yielding thaw

depths of around 0.3 m and less. On the first terrace, this effect is somewhat reduced (thinner and wetter organic top layer and lower water ice contents below), while the second terrace lacks the organic top layer and as a consequence experiences considerably deeper thawing than the two other stratigraphic units. In addition, the summer surface forcing strongly impacts thaw depths. Within the first terrace, the model results yield a pronounced north-south gradient of thaw depths (Fig. 13) which is related to the pattern of thawing degree days (Fig. 5).

## 5.4 Towards remote detection of ground temperature and thaw depth in permafrost areas?

The presented model approach can compute ground temperatures and thaw depths for an area of more than $10\,000\,km^2$, largely based on remotely sensed data sets. Other than in satellite-based approaches with much simpler steady-state models (Hachem et al., 2009; Westermann et al., 2015), the time evolution of the ground thermal regime is explicitly accounted for in the transient approach using CryoGrid 2. Our results suggest that the annual temperature amplitude to about 2 to 3 m depth is

generally captured, while a longer time series is needed to evaluate and secure multi-annual trends, in particular since the first part of the model period is affected by the initialization. However, with the ever extending record of high-quality satellite data, remote detection of trends in permafrost temperatures may become feasible within the coming years.

Sufficient computational resources provided, the presented scheme could in principle be extended to the entire Northern Hemisphere, for which GlobSnow retrievals are available. However, at present such application is limited by a number of shortcom-

ings and complications: first, the model scale of $1\,km^2$ may be sufficient to represent the ground thermal regime in lowland tundra landscapes like the LRD, but is significantly too coarse for heterogeneous terrain, e.g. in mountain areas (Fiddes et al.,

2015). Since the grid cell size is determined by the spatial resolution of the remotely sensed land surface temperatures, it could only be improved with the deployment of higher-resolution remote sensors for surface temperature (which must also feature a high temporal resolution). The snow density is a crucial parameter in the model scheme which has been determined from in-situ measurements in this study. For application on larger domains, spatial differences in snow density must be considered, which might be obtained e.g. from simple empirical relationships with climate variables (Onuchin and Burenina, 1996). Furthermore, remotely sensed data sets of snow water equivalent are lacking in many regions, in particular in coastal and mountain areas
(compare Fig. 5), and the spatial resolution of 25 km is hardly sufficient to capture the spatial distribution of snow in the terrain in complex landscapes. Furthermore, operational SWE retrievals are associated with considerable uncertainty in lake-rich tundra areas (Takala et al., 2011). In many permafrost areas, this can be expected to results in a strongly reduced accuracy so that significantly simpler schemes (Westermann et al., 2015) might provide similar results. Another crucial issue is the lack of a standardized pan-arctic product on subsurface properties, which combines spatially resolved classes with information on
subsurface stratigraphies and thermal properties. There exists a variety of such products on the regional and local scales, but they strongly differ in their quality and classes which are derived for different purposes. A pan-arctic homogenization effort similar to what has been accomplished for permafrost carbon stocks (Hugelius et al., 2013) is therefore needed in order to obtain meaningful results with a transient ground thermal model, such as CryoGrid 2.

Despite such challenges, transient ground temperature modeling forced by remote sensing data offers great prospects for
permafrost monitoring in remote areas that are not covered by in-situ measurements. The good performance regarding thaw depths and the timing of the seasonal thaw progression (Fig. 13) suggests that the results may even help estimating the release of greenhouse gases as a consequence of active layer deepening in a warming climate (Schuur et al., 2015).

## 6   Conclusions

We present a modeling approach to estimate the evolution of the ground thermal regime in permafrost areas at 1 km spatial and
weekly temporal resolution, based on a combination of satellite data and reanalysis products. The scheme is applied to an area of $16\,000\,km^2$ the Lena River Delta in Northeastern Siberia where measurements of ground temperatures and thaw depths are available to evaluate the performance. The approach is based on the 1D ground thermal model CryoGrid 2, which calculates the time evolution of the subsurface temperature field based on forcing data sets of surface temperature and snow depth for each grid. As forcing data, we synthesize weekly average surface temperatures from MODIS Land Surface Temperature
products and near-surface air temperatures from the ERA-interim reanalysis. For snow depth, low-resolution remotely sensed GlobSnow Snow Water Equivalent data are combined with higher-resolution satellite observations of snow extent facilitating an adequate representation of the snow start and end dates in the model. For the subsurface domain, a classification based on geomorphological mapping has been compiled, which can resolve the large-scale differences in e.g. ground-ice and soil-water contents. The model was subsequently run for a period of 14 years (2000-2014) and the results compared to observations of
the ground temperatures and thaw depths at nine sites.

- The forcing data sets in general agree well with multi-year in-situ observations. Monthly average surface temperatures are reproduced within $1°C$ or less, while the snow start and end dates in most years agree within one week. In a few years, larger deviations of up to three weeks occur.

- The comparison of model results to in-situ measurements suggests that the approach can reproduce the annual temperature amplitude. Multi-annual averages of ground temperatures at 2 to 3 m depth are reproduced with an accuracy of 1 to $2°C$, while comparison of monthly averages yielded an overall RMSE of $1.1°C$ and a cold-bias of $0.9°C$ for the model results. However, due to the small number of validation sites, this accuracy assessment must be considered preliminary.

- Modeled thaw depths in general agree with in-situ observations within 0.1 to 0.2 m. At one site, comparison with a multi-annual time series of thaw depth measurements suggests that the model scheme is capable of reproducing interannual differences in thaw depths with an accuracy of approx. 0.05 m.

- A sensitivity analysis showcases the influence of the subsurface stratigraphy on both ground temperatures and thaw depths, with temperature differences up to $2°C$ and thaw depth differences of a factor of three between classes for the same forcing data.

- The highest average ground temperatures are modeled for grid cells close to the main river channels and areas featuring sandy sediments with low organic contents in the northwestern part of the Lena River Delta. The lowest modeled ground temperatures occur in the eastern part of the delta towards the coastline, and in areas with ice-rich Yedoma sediments.

- The lowest thaw depths are modeled for Yedoma in the southern parts of the delta, as well as in areas with both low snow depths and cold summer surface temperatures in the Northeastern part. The deepest thaw depths are found in areas where the stratigraphy assigns mineral ground with low ice and organic contents.

The results of this study encourage further development of satellite-based modeling of the ground thermal regime in permafrost areas on continental scales. The largest obstacles are the lack of a standardized classification product on subsurface stratigraphies and thermal properties, as well as shortcomings and limitations of the currently available remote products on snow depth and snow water equivalent (see Sect. 5.4). If such limitations can be overcome, remote sensing-based methods could complement and support ground-based monitoring of the ground thermal regime.

*Acknowledgements.* This work was funded by SatPerm (project no. 239918; Research Council of Norway), COUP (project no. 244903/E10; JPI Climate; Research Council of Norway), Page21 (contract no. GA282700; European Union FP7-ENV), ESA GlobPermfrost (www.globpermafrost.info) and the Department of Geosciences, University of Oslo, Norway. We gratefully acknowledge the support of the Russian-German research station on Samoylov Island. We thank Günther Stoof, Sofia Antonova and Sascha Niemann for their contributions to fieldwork.

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

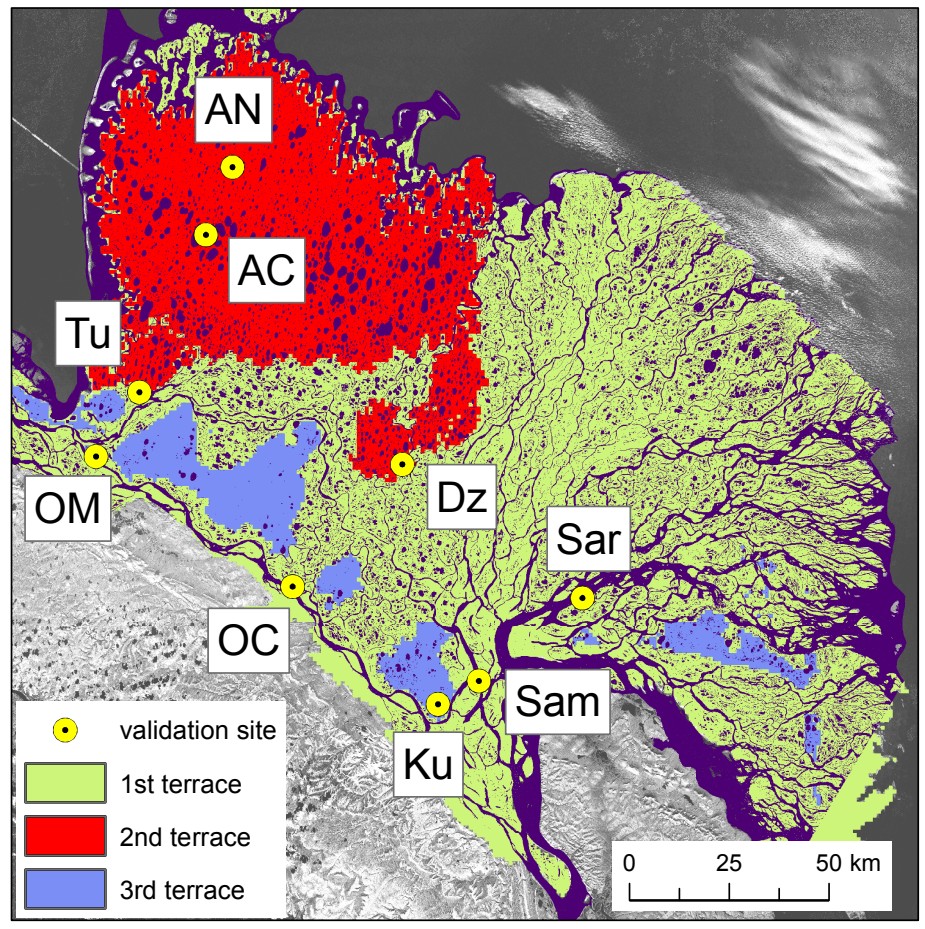

**Figure 1.** The Lena River Delta with the three stratigraphic classes distinguished in the ground thermal modeling (Sect. 3.2) and sites with in-situ observations (Sect. 2.2.2) employed for model validation. AN: Arga Island, north; AC: Arga Island, center; Dz: Dzhipperies Island; Ku: Kurungnakh Island; OC: Olenyokskaya Channel, center; OM: Olenyokskaya Channel, mouth; Sam: Samoylov Island; Sar: Sardakh Island; Tu: Turakh Island.

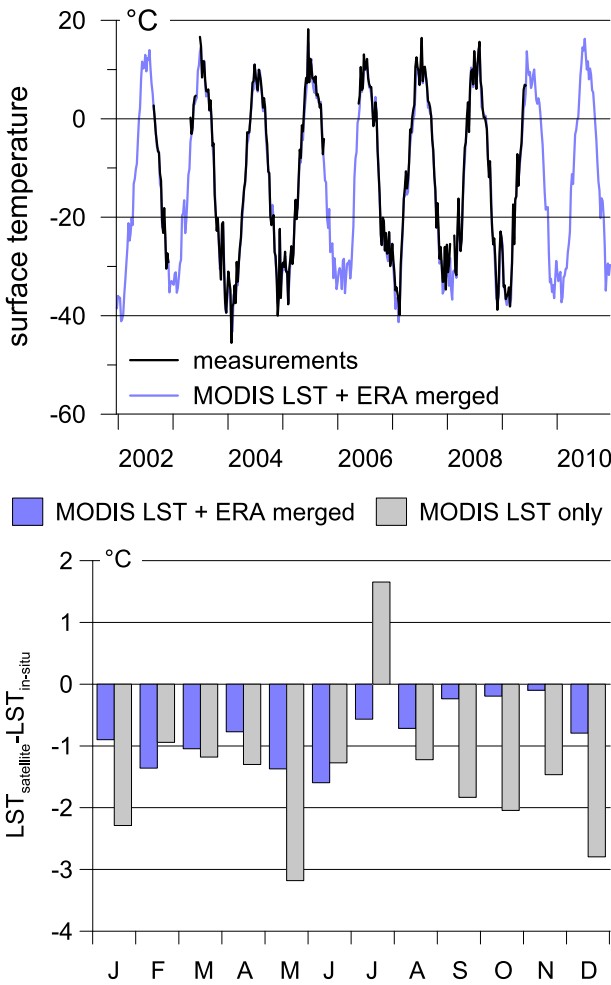

**Figure 2.** Top: daily average surface temperatures measured on Samoylov Island (Langer et al., 2013; Boike et al., 2013) vs. surface temperatures synthesized from MODIS LST and ERA reanalysis. Bottom: difference between satellite-derived LST and in-situ measurements for monthly averages of periods when in-situ measurements are available (see top figure). See text.

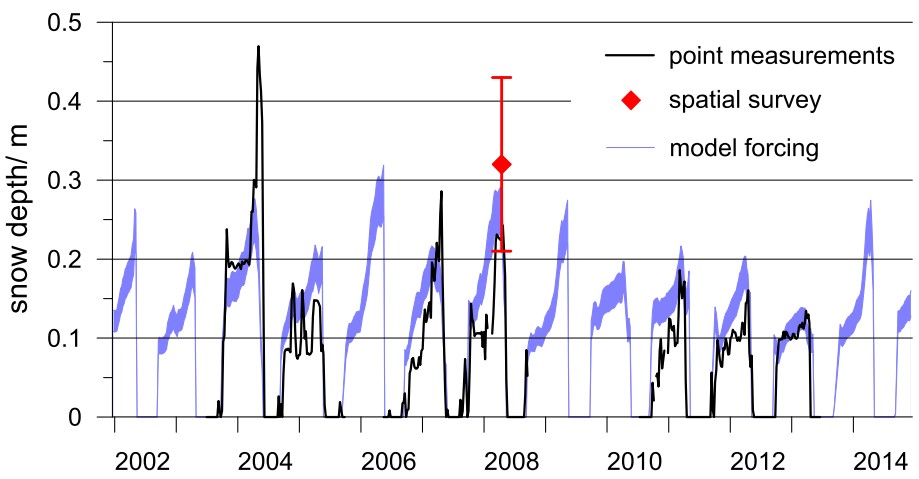

**Figure 3.** Modeled and measured snow depths on Samoylov Island (Boike et al., 2013). The point measurements are conducted with an ultrasonic ranging sensor (data smoothed with running average filter with window size of one week, corresponding to the temporal resolution of the model forcing), the spatial survey is based on manual measurements at 216 points in polygonal tundra conducted between 25 April and 2 May 2008 (Fig. 6a, Boike et al., 2013). The blue area depicts the spread between model runs with snow densities of 200 and 250 kg m$^{-3}$.

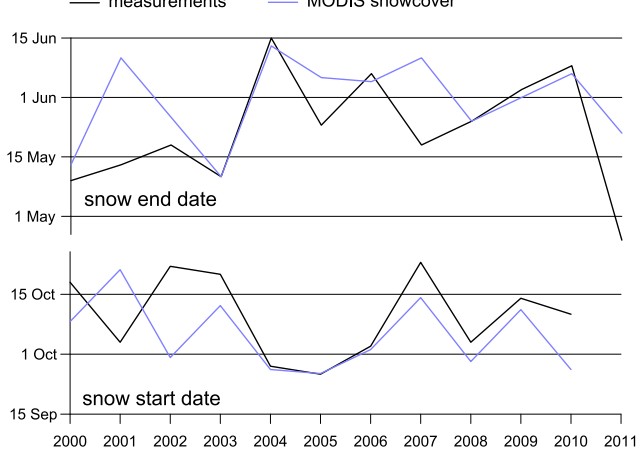

**Figure 4.** Modeled and measured snow start and end on Samoylov Island (Boike et al., 2013).

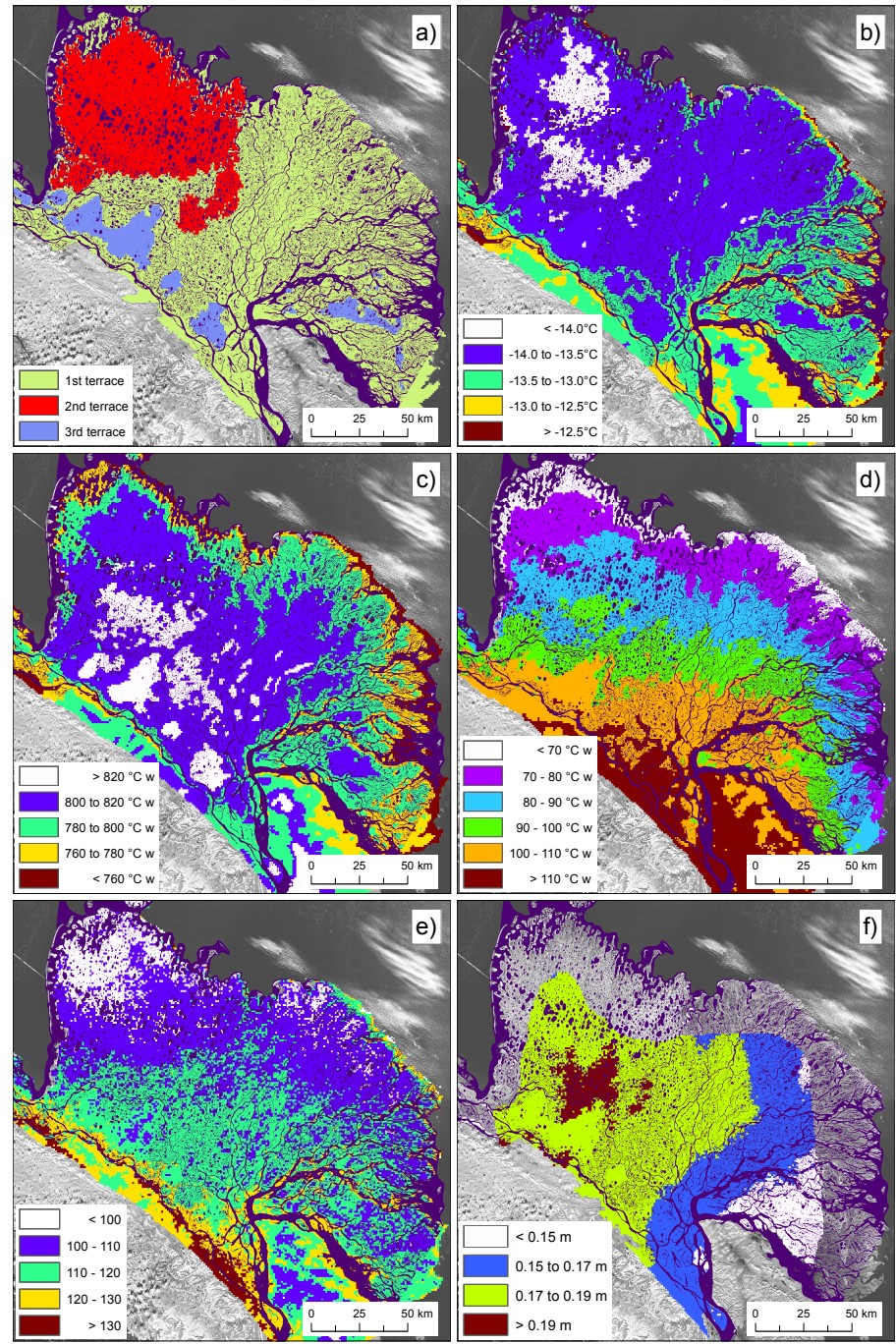

**Figure 5.** Spatial distribution of model input data sets in the LRD (Sects. 3.2, 3.3): a) subsurface classification (compare Table 1); b) average surface temperature 2004-2013; c) average freezing degree weeks 2004-2013; d) average thawing degree weeks 2004-2013; e) average number of snow-free days 2004-2013; f) average snow depth 2004-2013 for a snow density of 225kg m$^{-3}$.

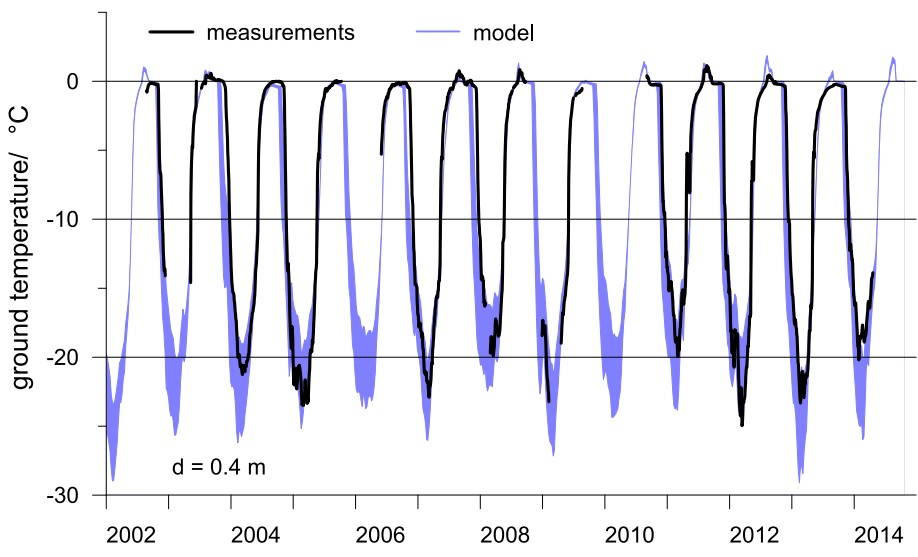

**Figure 6.** Modeled and measured ground temperatures at a depth of $0.4\,\mathrm{m}$ at a wet polygon center on Samoylov Island (Boike et al., 2013). The blue area depicts the spread between model runs with snow densities of 200 and $250\,\mathrm{kg\,m^{-3}}$. The temperature sensor drifted by about $-0.2\,^{\circ}\mathrm{C}$ (at $0\,^{\circ}\mathrm{C}$) in the shown period.

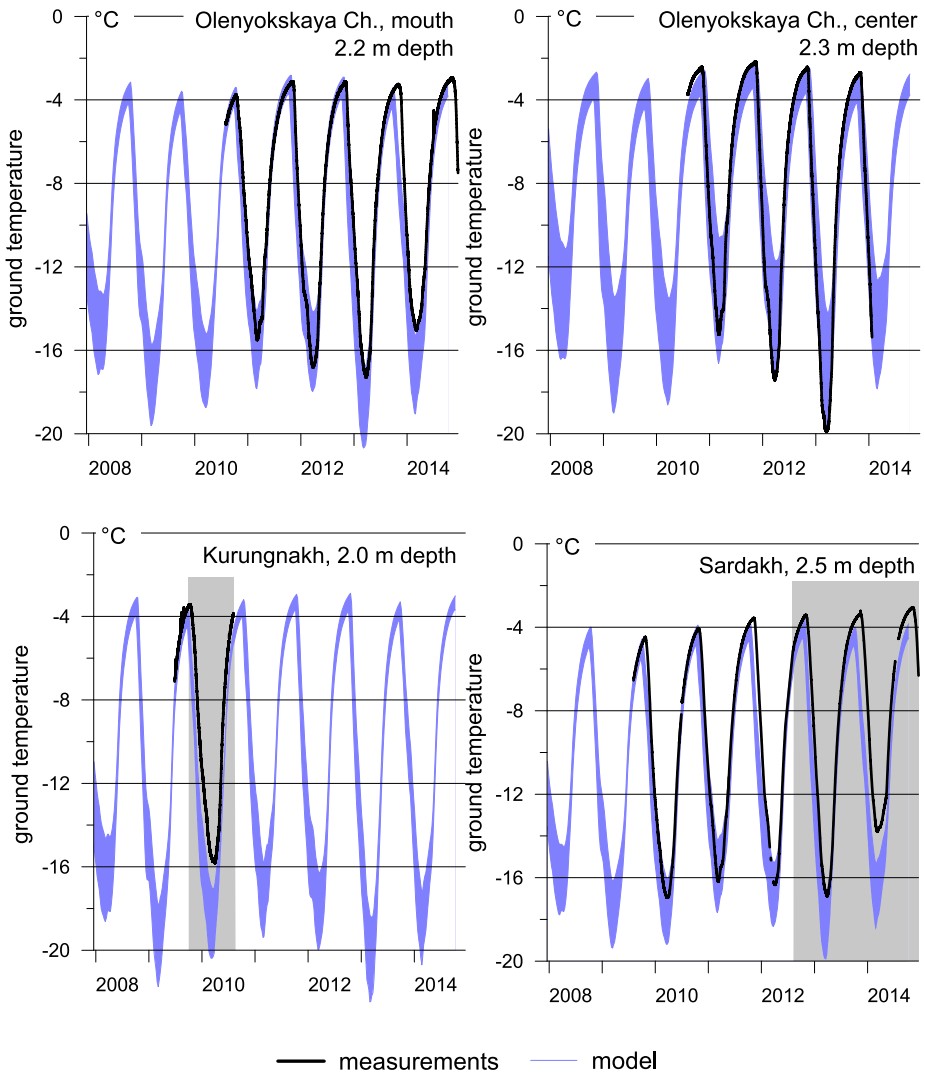

**Figure 7.** Modeled and measured ground temperatures at depths of 2.0-2.5 m at four locations in the LRD. The blue area depicts the spread between model runs with snow densities of 200 and $250\,\mathrm{kg\,m^{-3}}$. Periods for which in-situ data are affected by thermokarst are marked in grey. These should not be used for comparison, see text.

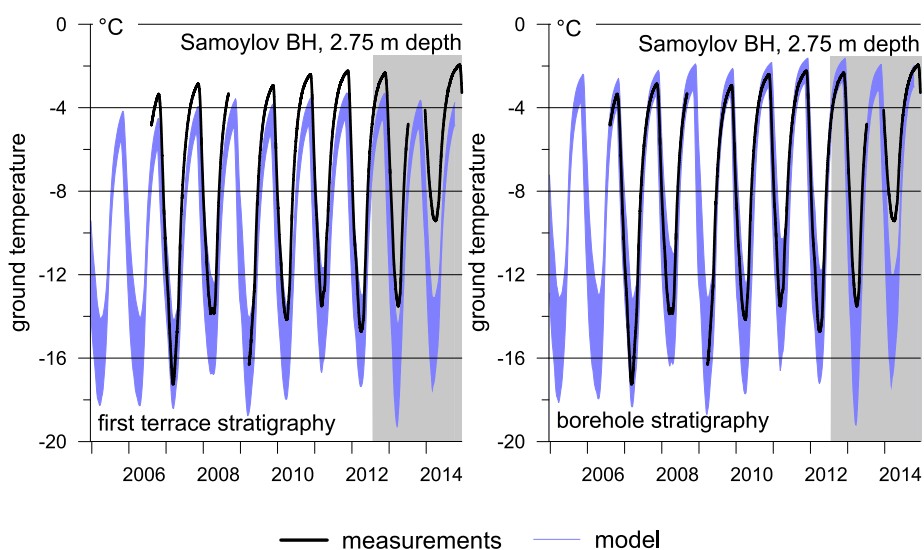

**Figure 8.** Modeled and measured ground temperatures for the borehole on Samoylov Island. Left: subsurface stratigraphy of the first terrace (Table 1). Right: stratigraphy adapted to the true ground conditions at the borehole (0-0.5 m: 30% water/ice, 10% air, 60% mineral, sand; 0.5-9 m: 40% water/ice, 60% mineral, sand; deeper layers as for first terrace, Sect. 4.2.1). The blue area depicts the spread between model runs with snow densities of 200 and $250 \, \mathrm{kg \, m^{-3}}$. Periods for which in-situ data are affected by new installations at the Samoylov station are marked in grey. These should not be used for comparison, see text.

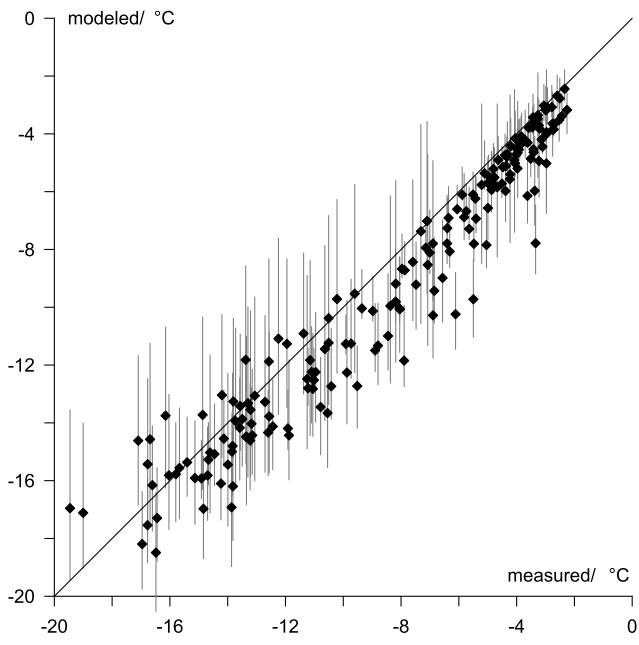

**Figure 9.** Modeled and measured monthly average ground temperatures for the LRD boreholes and 1:1 line (n=185, data as shown in Figs. 7 and 8 right). Olenyokskaya Channel mouth and center: full time series; Kurungnakh Island: time series until September 2009; Samoylov Island: time series until August 2012, model data with borehole stratigraphy (Fig. 8 right); Sardakh Island: time series until August 2012. Vertical bars: spread between model runs with snow densities of 200 and $250\,\mathrm{kg\,m^{-3}}$; diamonds: model run with snow density $225\,\mathrm{kg\,m^{-3}}$.

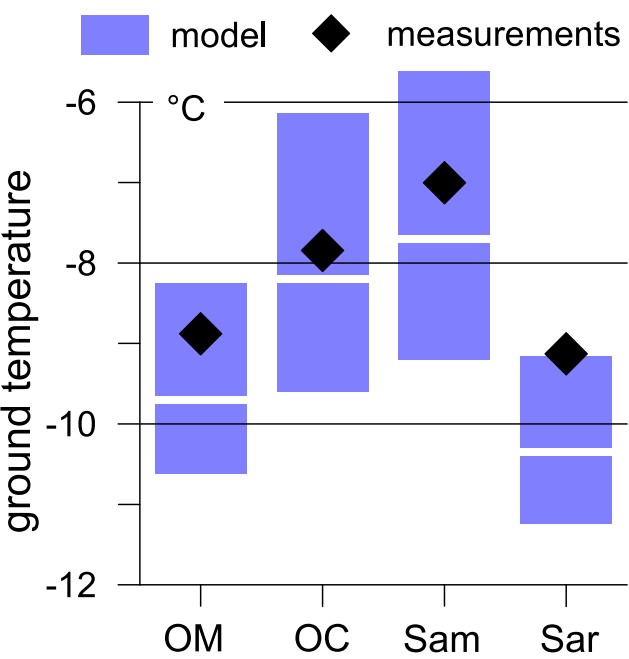

**Figure 10.** Modeled and measured annual average ground temperatures for the LRD boreholes for the two-year period September 2010 to August 2012 (OM: Olenyokskaya Channel mouth; OC: Olenyokskaya Channel center; Sam: Samoylov Island borehole; Sar: Sardakh Island). Blue bar: spread between model runs with snow densities of 200 and 250 $\mathrm{kg\,m^{-3}}$; white line: model run with snow density 225 $\mathrm{kg\,m^{-3}}$. The ground temperatures correspond to the depths given in Figs. 7 and 8, for Samoylov, the simulations for the borehole stratigraphy (Sect. 4.2.1, Fig. 8 right) are presented.

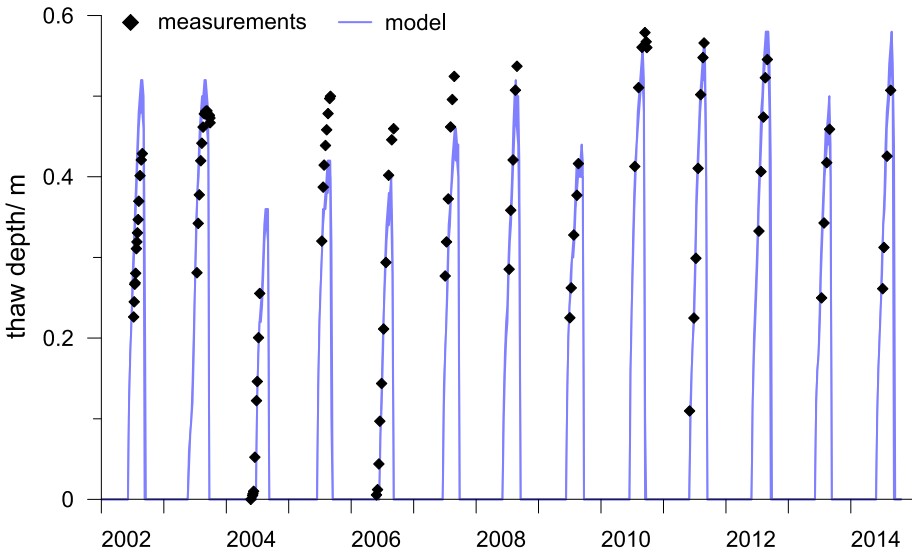

**Figure 11.** Modeled and measured thaw depths on Samoylov Island. The measurements correspond to the average of 150 locations on Samoylov Island (Boike et al., 2013). The average standard deviation of the measurements (i.e. the spatial variability of thaw depths) is 0.06 m. The blue area depicts the spread between model runs with snow densities of 200 and $250\,\mathrm{kg\,m^{-3}}$.

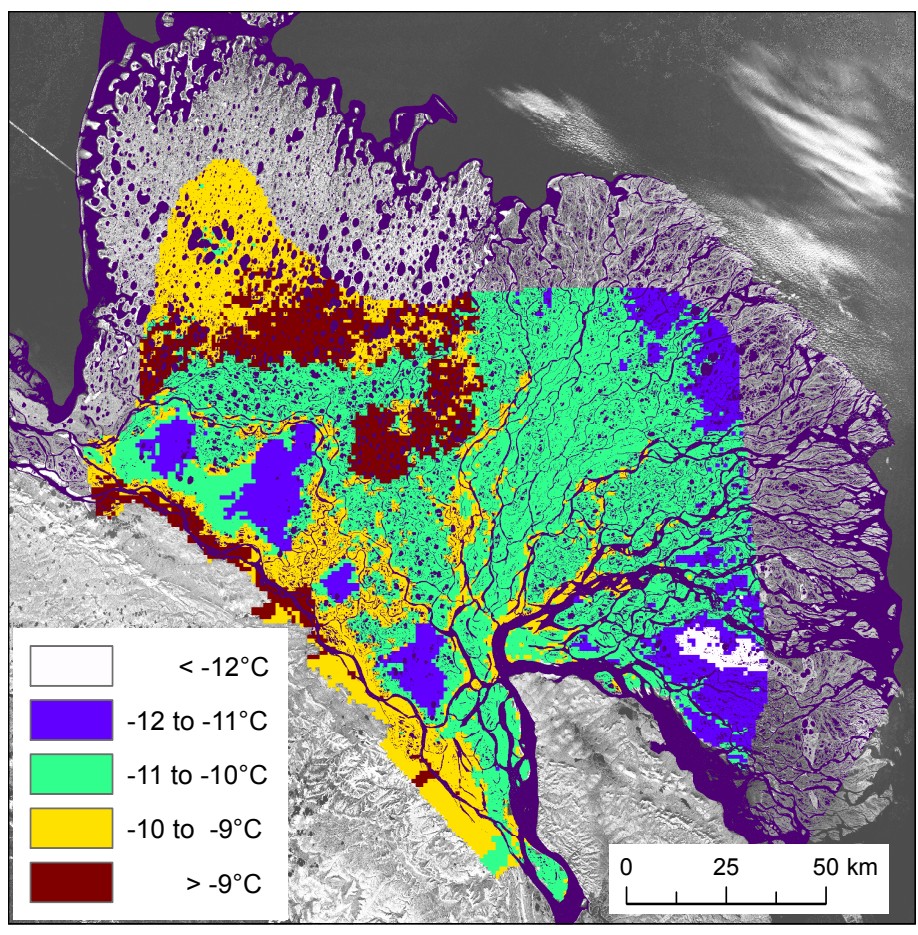

**Figure 12.** Modeled average ground temperatures at 1 m depth for the period 2004-2013, with a snow density of $225\,\mathrm{kg\,m^{-3}}$.

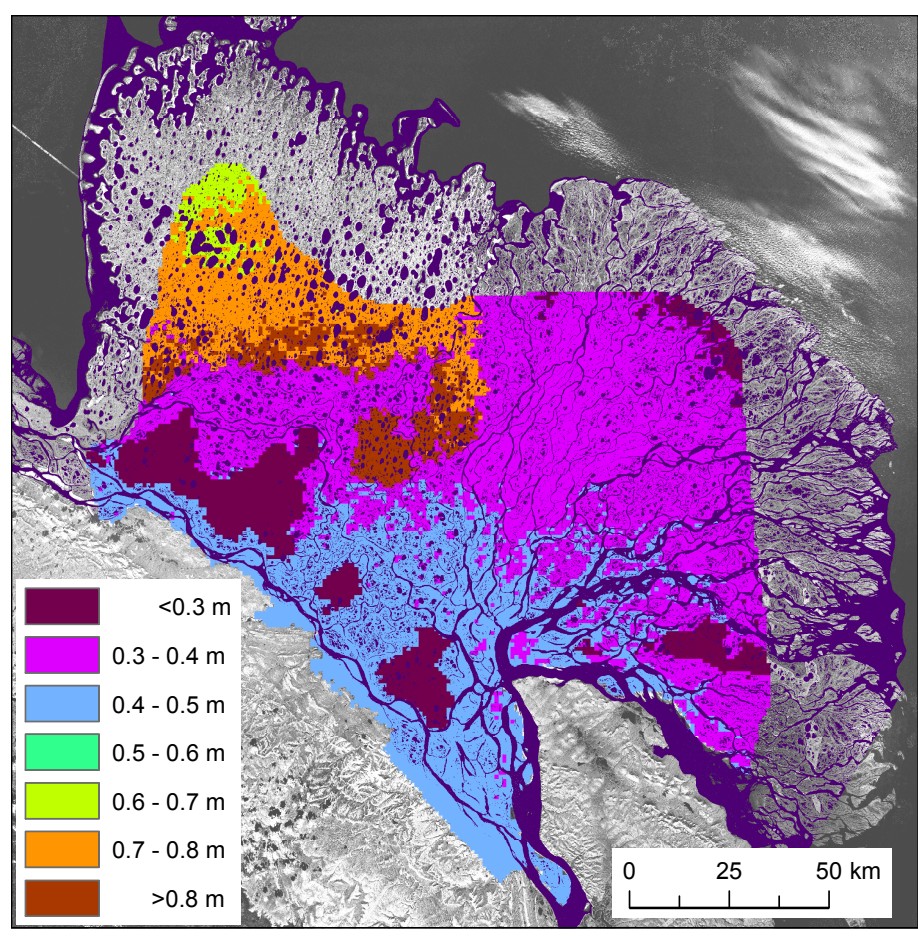

**Figure 13.** Modeled average maximum thaw depths for the period 2004-2013, with a snow density of $225\,\mathrm{kg}\,\mathrm{m}^{-3}$.

**Table 1.** Subsurface stratigraphies for the three LRD terraces with volumetric fractions of the soil constituents and sediment type assigned to each layer.

| depth [m] | water/ice | mineral | organic | air | type |
|---|---|---|---|---|---|
| **First Terrace** | | | | | |
| 0–0.15 | 0.6 | 0.1 | 0.15 | 0.15 | sand |
| 0.15–9 | 0.65 | 0.3 | 0.05 | 0.0 | silt |
| >9 | 0.3 | 0.7 | 0.0 | 0.0 | sand |
| **Second Terrace** | | | | | |
| 0-10 | 0.4 | 0.6 | 0.0 | 0.0 | sand |
| >10 | 0.3 | 0.7 | 0.0 | 0.0 | sand |
| **Third Terrace - Yedoma** | | | | | |
| 0–0.15 | 0.3 | 0.1 | 0.1 | 0.5 | sand |
| 0.15–20 | 0.7 | 0.25 | 0.05 | 0.0 | sand |
| >20 | 0.3 | 0.7 | 0.0 | 0.0 | sand |

**Table 2.** Modeled and measured thaw depths in the LRD for confining snow depths of $200 \mathrm{kg\,m^{-3}}$ and $250 \mathrm{kg\,m^{-3}}$.

| Site | date | measured | modeled | |
|---|---|---|---|---|
| | | | $200 \mathrm{kg\,m^{-3}}$ | $250 \mathrm{kg\,m^{-3}}$ |
| Samoylov Island | 2002-2014 | see Fig. 11 for detailed comparison | | |
| Olenyokskaya Ch., center | 16 Aug 2010 | 0.6 m | 0.55 m | 0.51 m |
| Arga Island, North | 11 Aug 2010 | 0.9-1.0 m | 0.84 m | 0.80 m |
| Arga Island, Center | 3 Aug 1998 | 0.6 m | 0.61 m | 0.60 m |
| | | | average 3 Aug, 2001-2010 | |
| Dzhipperies Island | 23 Jul 1998 | 0.7 m | 0.68 m | 0.64 m |
| | | | average 23 Jul, 2001-2010 | |
| Turakh Island | 20-29 Aug 2005 | 1.0-1.1 m | 0.74 m | 0.70 m |
| Olenyokskaya Ch., mouth | 14 Aug 2010 | 0.2 m | 0.29 m | 0.27 m |
| Kurungnakh Island | 14/15 Jul 2013 | 0.12-0.18 m | 0.19-0.20 m | 0.19-0.20 m |
| (9 sites, | 9/10 Aug 2013 | 0.16-0.22 m | 0.26-0.28 m | 0.20-0.21 m |
| 6 grid cells) | 26 Aug 2013 | 0.21-0.26 m | 0.29-0.30 m | 0.28-0.29 m |

**Table 3.** Sensitivity of modeled average ground temperatures at 1 m depth and average maximum thaw depth over the period 2004-2013. All simulations with snow density 225 $\text{kg m}^{-3}$.

| Site | ground temperature/ $^\circ$C | | | thaw depth/ m | | |
|------|------|------|------|------|------|------|
| | 1st | 2nd | 3rd | 1st | 2nd | 3rd |
| | terrace stratigraphy | | | terrace stratigraphy | | |
| Arga Island, north | -11.6 | -10.3 | -12.2 | 0.30 | 0.69 | 0.19 |
| Arga Island, center | -11.3 | -10.0 | -12.1 | 0.30 | 0.71 | 0.19 |
| Dzhipperies Island | -10.6 | -9.0 | -11.5 | 0.39 | 0.86 | 0.24 |
| Kurungnakh Island | -10.6 | -9.0 | -11.5 | 0.46 | 0.96 | 0.28 |
| Olenyokskaya Ch., mouth | -9.7 | -8.0 | -10.8 | 0.43 | 0.93 | 0.26 |
| Olenyokskaya Ch., center | -9.5 | -7.9 | -10.6 | 0.45 | 0.96 | 0.28 |
| Samoylov Island | -10.2 | -8.6 | -11.1 | 0.46 | 0.97 | 0.28 |
| Sardakh Island | -10.5 | -9.0 | -11.3 | 0.41 | 0.90 | 0.25 |
| Turakh Island | -10.7 | -9.2 | -11.6 | 0.38 | 0.94 | 0.22 |