# Peer review of "Transient modeling of the ground thermal conditions using satellite data in the Lena River Delta, Siberia"

_The Cryosphere, 2016_

## Referee Comment (RC1) · Anonymous Referee #1 · 5 Aug 2016

In this study the authors present a remote sensing based scheme for transient modelling of the ground surface regime together with the previously published numerical model CryoGrid2. The scheme is applied over a large area in the Lena River Delta (LRD), Siberia. Forcing datasets at 1km and weekly resolution are derived from MODIS LST, MODIS SCE, GlobSnow SWE plus meteo fields from ERA-Interim reanalysis. Spatially distributed ground properties are based on geomorphological observations and mapping drawing on previous studies in the region. Results are compared to in-situ observations of ground temperatures from boreholes, CALM active layer depths and measurements from the Samoylov Island Permafrost observatory. The authors conclude that comparison to in-situ measurements shows that the scheme is capable

of estimating the thermal state of permafrost and its time evolution in the LRD.

This paper is a further contribution to the work of using remote sensing data together with numerical models (eg. Westermann 2015) which I think is a very interesting and promising approach to large area and/or operational assessments. The paper is well written with a clear methodology, presentation of results and critical discussion. The authors acknowledge shortcomings of the approach such as dependency upon a well estimated snow density and difficulty in applying in heterogeneous terrain due to coarse scale of the LST data. I have one main comment with respect to the forcing timeseries, other comments are reasonably minor.

Comments:

1. P8 Section 3.3: In the merged LST /reanalysis product, 2m air temperature and LST are merged. I think it would be helpful to add some discussion of how comparable surface air temperature and LST are and how this is expected to vary under both different atmospheric and surface cover conditions. The most obvious example is when a snowcover is present and air temperature and snow surface temperature can differ strongly. This reference (Gallo et al 2011) would probably be useful: http://dx.doi.org/10.1175/2010JAMC2460.1. This study from Raleigh et al. http://dx.doi.org/10.1002/2013WR013958 suggests that the 2metre dewpoint temp (also available from ERA-Interim) is perhaps a better approximation of snow surface temperature than 2m air temperature. What kind of biases can be expected by forcing the upper boundary condition of surface temperature with a 2m air temperature field? Or are these different forcings treated differently by the model?

2. P10 l20-22: is this spatial variability due to residual snow patches? Perhaps state the cause here.

3. P7 l3 + 33 on line 3 you say "extensive set of observations available" whereas on l33 you say "which temporally /spatially distributed sets are not available" - are these statements contradictory? Can you describe the snow density data briefly in Section

2.2, particularly at which times of year these measurements were made.

4. Fig 6: Is there an offset in your measurements as looks like in Fig 6 that zero curtain is occurring 0.5deg or so below the 0degC point.

5. Fig 6: can you explain why there is no zero curtain at phase transition from ice to water in spring/summer in the wet polygon? Would you expect this?

Technical issues:

1. p6 l29: add terms in brackets after items in text so that equation is more easily understood.

2. P7 l6: ...LRD for which... → ...LRD which..

3. P9 l27-29: I think it is more common to use term "layers" when talking about vertical discretisation of model units?

4. P10 l16: Figure 2 seems to lose most bar elements upon printing (not digital form). Perhaps my printer issue - but check this.

5. P10 l24: "well suited as input for ground thermal modelling" - qualify this statement with something like ", at least in homogeneous terrain".

6. P11 l7 over a an → over an.

7. p11 l10 repeated word "cloudiness" → you mean snow?

8. P14 l32-33: qualify statement with something like 'in homogeneous terrain'.

9. P17 l11: had → hand.

10. p18 l8: ares → area.

---

## Referee Comment (RC2) · Anonymous Referee #2 · 17 Oct 2016

This paper presents an approach to map the spatial distribution of ground temperatures and thaw depths using a 1D transient ground thermal model (CryoGrid 2). The model uses remote sensing derived surface temperature (MODIS 1 km complemented by 2-m air temperature from atmospheric reanalysis ERA-Interim 0.75 deg. grid spacing) and snow depth obtained from the GlobSnow snow water equivalent (SWE; 25 km grid spacing) product as forcing data. The study builds on the earlier work of Langer et al. (2013), moving the application of the CryoGrid 2 model from the local scale (station on Samoylov Island) to the regional scale by including the entire Lena River Delta (LRD). From Figure 1, one notices that the LRD is covered by many small lakes and branches of the Lena River (i.e. a large freshwater fraction). This is a complex area to study using coarse resolution passive microwave satellite data (or derived products) due to the large sub-grid scale variability within pixels, notably due to the presence of water bodies, which introduces significant uncertainty in SWE estimates (GlobSnow or other satellite-based SWE products). This issue has been recognized by the group who originally developed the GlobSnow product (reported in Takala et al., 2011) and at least one of its members in a latter publication (Derksen et al., 2012). Takala et al. (2011; Table 3) report mean bias errors of -36 mm and RMSE of 47 mm for a tundra area with small water bodies. Derksen et al. (2012) also show that SWE retrieval errors can be large (see Table 9 and Figure 12 of this publication) from passive microwave data, even more revealing (over 100% error; Figure 12) when examining errors on a monthly basis. As indicated by Takala et al. (2011): "Additionally, the consideration or compensation for the effect of (frozen) lakes requires further study and algorithm development work."

The authors of the present manuscript state, in Langer et al. (2013), that: "**The thermal state of permafrost is reproduced with an uncertainty of about ±2.5 °C with a SWE accuracy of about ±10 mm.** This is still below the performance that can be reached with a realistic LST accuracy of about ±2 °C. **However, a much lower SWE accuracy level (±40 mm) must be considered in regions with sparse weather stations (Luojus et al., 2010) and when field measurements are not available for calibration. Our results show that realistic permafrost simulations with a transient heat transfer model would be almost impossible with such low accuracies in the SWE forcing.** In contrast to the permafrost temperatures, the thaw depths are found to be more or less independent from the SWE accuracy. However, this might be different in regions where the permafrost temperature is already close to the freezing point as observed by Åkerman & Johansson (2008). In any case, the impact of snow on the active layer dynamics can be very complex and dependent on regional factors (Zhang, 2005). The performed sensitivity study demonstrates that a highly accurate snow cover forcing is crucial for reliable permafrost modeling."

Given: 1) the above statement by the authors in a previous paper; 2) the known retrieval errors in similar regions reported by the developers of GlobSnow SWE; and 3) the lack of validation of snow depth (derived from GlobSnow SWE with density values of 200-250 kg m$^{-3}$) over a larger area (transects) than just the small island of Samoylov (located to the south of the LRD), I am afraid to say that the manuscript submitted is not acceptable for publication in *The Cryosphere*. In fact, I am quite concerned by the fact that the authors missed the publication of Takala et al. (2011) which is the key paper reporting uncertainties of the GlobSnow SWE product. It is important to read and cite others who work in similar areas or at least with similar data sets, and who have reported uncertainties in the forcing variables used by CryoGrid 2.

Other remarks:

1. The authors do not seem to be aware that the SSM/I footprints for the 19 GHz and 37 GHz frequency brightness temperature channels are in the order of 70x45 km and 38x30 km, respectively. These brightness temperature measurements are then interpolated into a 25x25 km grid which is then used for SWE retrieval in GlobSnow. Therefore, although the authors masked some areas along the coast, the ocean "overspill" problem within the footprints is a larger problem than reported herein.
2. The large fraction of the landscape covered by lakes/river channels represents the largest uncertainty in GlobSnow SWE values. The authors need to read further on this topic in order to better understand the limitations of GlobSnow SWE and, perhaps, search for other products (satellite or reanalysis, including assemble) that could be considered in a new manuscript submission to *The Cryosphere* or another journal.
3. Boike et al. (2013) is given as the reference for the snow depth and density values of Samoylov Island. However, I personally browsed this paper to find that there are mismatches between values reported in Table 5 and Figure 6 of that paper and the values reported in Figure 3 (and the text) of the present manuscript. I am not sure how, as a reviewer, I can reconcile the two sources. The range (and maximum) of measured snow depths in Boike et al. (2013) do not always match those of this paper. For example, in winter 2004 (a high snow year), the maximum snow depth found in Table 5 and Figure 6 of Boike et al. (2013) is 56 cm while that plotted in the graph of Figure 3 of this paper is at a value of about 47 cm. This is only one of several examples.
4. How much confidence should we have in the snow depth map of Figure 5 and the ground temperature (1-m depth) map of Figure 11, given that snow density comes from Samoylov Island only and that there is a large degree of uncertainty in GlobSnow SWE retrievals over complex (lake-rich) areas such as the LRD? As shown in Figures 6-8, winter temperatures are significantly underestimated in wintertime by the model (up to $8\,^{o}$C, most frequently by 3-4$^{o}$C). Of course, taking the average of all years combined reduces the error reported (1-1.5$^{o}$C given in the Abstract), but the errors are larger when inspecting each individual year.
5. The scaling issue between point (single station measurement(s)) and large satellite pixels should not be ignored throughout the manuscript.

Additional references not included in manuscript:

Derksen, C., Toose, P., Lemmetyinen, J., Pulliainen, J., Langlois, A., Rutter, N., & Fuller, M.C. (2012). Evaluation of passive microwave brightness temperature simulations and snow water equivalent retrievals through a winter season. *Remote Sensing of Environment*, 117, 236–248.

Takala, M., Luojus, K., Pulliainen, J., Derksen, C., Lemmetyinen, J., Kärnä, J. -P., Koskinen, J., & Bojkov, B. (2011). Estimating northern hemisphere snow water equivalent for climate research through assimilation of space-borne radiometer data and ground-based measurements. *Remote Sensing of Environment*, doi:10.1016/j.rse.2011.08.014.

---

## Author Comment (AC1) · 12 Dec 2016

**Reply to Reviewer 1**

We are grateful to the reviewer for the thoughtful comments and suggestions to our manuscript. We have compiled a revised version and in the following provide a point-by-point reply to all issues raised. The reviewer's comments appear in bold font, our replies in normal font, and changes to the manuscript are in italics.

In this study the authors present a remote sensing based scheme for transient modelling of the ground surface regime together with the previously published numerical model CryoGrid2. The scheme is applied over a large area in the Lena River Delta (LRD), Siberia. Forcing datasets at 1km and weekly resolution are derived from MODIS LST, MODIS SCE, GlobSnow SWE plus meteo fields from ERA-Interim reanalysis. Spatially distributed ground properties are based on geomorphological observations and mapping drawing on previous studies in the region. Results are compared to insitu observations of ground temperatures from boreholes, CALM active layer depths and measurements from the Samoylov Island Permafrost observatory. The authors conclude that comparison to in-situ measurements shows that the scheme is capable of estimating the thermal state of permafrost and its time evolution in the LRD.

This paper is a further contribution to the work of using remote sensing data together with numerical models (eg. Westermann 2015) which I think is a very interesting and promising approach to large area and/or operational assessments. The paper is well written with a clear methodology, presentation of results and critical discussion. The authors acknowledge shortcomings of the approach such as dependency upon a well estimated snow density and difficulty in applying in heterogeneous terrain due to coarse scale of the LST data. I have one main comment with respect to the forcing timeseries, other comments are reasonably minor.

**Comments:**

1. P8 Section 3.3: In the merged LST /reanalysis product, 2m air temperature and LST are merged. I think it would be helpful to add some discussion of how comparable surface air temperature and LST are and how this is expected to vary under both different atmospheric and surface cover conditions. The most obvious example is when a snowcover is present and air temperature and snow surface temperature can differ strongly. This reference (Gallo et al 2011) would probably be useful: http://dx.doi.org/10.1175/2010JAMC2460.1. This study from Raleigh et al. http://dx.doi.org/10.1002/2013WR013958 suggests that the 2metre dewpoint temp (also available from ERA-Interim) is perhaps a better approximation of snow surface

**temperature than 2m air temperature. What kind of biases can be expected by forcing the upper boundary condition of surface temperature with a 2m air temperature field? Or are these different forcings treated differently by the model?**

We have taken up this comment in Sect. 3.3 and in the Discussion, Sect. 5.1.1. In Sect. 3.3, we have explained how we handle situations when positive temperatures of the surface forcing (which can occur as a result of admixing of air temperatures) occur for still snow-covered ground:

"During cloudy skies, differences between air and surface temperatures are strongly reduced compared to clear-sky conditions (Gallo et al., 2011), so that air temperatures can be regarded an adequate proxy when MODIS LST is not available due to cloud cover. For melting snow, surface temperatures are confined to the melting point of ice, while air temperatures can be positive. Positive values of the surface temperature forcing are therefore set to 0°C when a snow cover is present."

In-situ measurements on Samoylov Island indicate that air temperatures are a relatively good proxy for snow surface temperatures in winter, most likely because the ground heat flux from the refreezing active layer and the cooling permafrost is a substantial source of energy to the surface which prevents strong near-surface temperature inversions. In Sect. 5.1.1 we write:

"Based on in-situ measurements, Raleigh et al. (2013) suggest that for snow-covered ground dew point temperatures are a better approximation for surface temperatures compared to air temperatures at standard height. However, observations on Samoylov Island suggest only a small offset between snow surface and air temperatures, with the difference increasing from near zero in early winter to about 1° C in late winter (Table 3, Langer et al., 2011b). The reason for this is most likely that the ground heat flux is a strong heat source especially in early winter (Langer et al., 2011b) which warms the surface and thus prevents formation of a strong nearsurface inversion. Therefore, we consider air temperatures an adequate proxy for snow surface temperatures in the LRD, but dew point temperatures should clearly be considered for gapfilling in the snow-covered season in future studies."

**2. P10 120-22: is this spatial variability due to residual snow patches? Perhaps state the cause here.**

In July, residual snow patches do not occur on Samoylov Island, the snow pack has fully melted by at latest mid June – the spatial variability is caused by different surface cover and soil moisture conditions. Using a thermal camera, Langer et al. (2010) showed that the spatial differences in polygonal tundra can be up to 10K for single scenes, but become much smaller for temporal averages over longer periods. However, a residual net difference between the point measurements used for comparison and the larger-scale MODIS LST values cannot be excluded. Text changed to: "However, surface temperatures can feature a strong spatial variability during summer due to differences in surface cover and soil moisture conditions..."

**3. P7 13 + 33 on line 3 you say "extensive set of observations available" whereas on 133 you say "which temporally /spatially distributed sets are not available" - are these statements contradictory?**

An extensive data set is available from Samoylov Island, but there is no data set covering the entire Lena River Delta (which we refer to in the second statement). We have made this clearer in the text, 1. 33 now reads:

"Therefore, the snow ... is a highly crucial parameter for which spatially or temporally distributed data sets covering the entire LRD are not available. However, an extensive set of measurements from polygonal tundra on Samoylov Island suggests ..."

**Can you describe the snow density data briefly in Section 2.2, particularly at which times of year these measurements were made.**

We have inserted a statement in Sect. 2.2:

"In addition, a spatially distributed survey of snow depths and densities (216 points in polygonal tundra) was conducted in early spring 2008 (25 April to 2 May) before the onset of snowmelt (Boike et al., 2013)."

In addition, the range of snow depths obtained from the spatial survey has been added to Fig. 3.

**4. Fig 6: Is there an offset in your measurements as looks like in Fig 6 that zero curtain is occurring 0.5deg or so below the 0degC point.**

Yes, there is a slight drift of the sensor in the later years (visible from 2010), with the zero curtain occurring at about -0.2°C instead of 0°C. A statement has been added to the figure caption.

**5. Fig 6: can you explain why there is no zero curtain at phase transition from ice to water in spring/summer in the wet polygon? Would you expect this?**

The zero curtain effect is a result of a two-sided freezing front which only occurs in fall for permafrost ground (the top freezes in fall and the ground below the active layer is permanently frozen). In this case, the temperatures at both sides of the still unfrozen domain are confined to  $0^{\circ}$ C due to the freezing of soil water at the freeze front. As a consequence, temperatures inside this domain quickly reach  $0^{\circ}$ C as well. As the freeze fronts progress slowly due to the considerable amounts of latent heat provided by the phase change of the water, this zero curtain state can last for several weeks and only ends when the two freeze fronts meet each other, which is generally followed by rapid cooling of the then frozen soil. In spring, the ground thaws from

top down and only one freeze/thaw front exists. In this case, there is always a temperature gradient both above and below the progressing thaw front, so that temperatures in a certain depth/grid cell are never confined to  $0^{\circ}$ C for extended periods. For seasonally frozen ground, the situation is opposite and the zero curtain effect occurs in spring, when the seasonally frozen layer thaws from top down and from the bottom up, again resulting in two freeze fronts. In fall, only on freeze front exists, corresponding to the freezing of the ground from top down, so that no zero curtain effect occurs.

**Technical issues:**

**1.** p6 l29: add terms in brackets after items in text so that equation is more easily understood.

done

2. P7 l6: ...LRD for which...  $\rightarrow$  ...LRD which...

Changed to "for which we define"

**3.** P9 127-29: I think it is more common to use term "layers" when talking about vertical discretisation of model units?

Changed to "layers"

4. P10 l16: Figure 2 seems to lose most bar elements upon printing (not digital form). Perhaps my printer issue - but check this.

We have tried printing Fig. 2, but did not encounter any problems.

5. P10 l24: "well suited as input for ground thermal modelling" - qualify this statement with something like ", at least in homogeneous terrain".

"at least in homogeneous terrain" added

**6. P11 l7 over a an $\rightarrow$ over an.**

Done, thanks!

7. p11 l10 repeated word "cloudiness"  $\rightarrow$  you mean snow?

Yes, we mean snow cover. Thanks!

**8. P14 l32-33: qualify statement with something like 'in homogeneous terrain'.**

We agree, this statement is too general. In the revised version, we have made clear that it only applies to our study area in the LRD, and only to homogeneous terrain. The sentence now reads:

"We conclude that surface temperatures synthesized from MODIS LST and ERA-interim reanalysis are an adequate choice for the purpose of ground thermal modeling in the LRD, at least in homogeneous terrain. However, it may introduce a slight cold-bias in modeled ground temperatures."

**9. P17 l11: had $\rightarrow$ hand.**

Done, thanks!

**10.** p18 l8: ares  $\rightarrow$  area.

Done, thanks!

New reference:

Langer, M., Westermann, S., Muster, S., Piel, K., and Boike, J.: The surface energy balance of a polygonal tundra site in northern Siberia Part 2: Winter, Cryosphere, 5, 509–524, 2011b.

---

## Author Comment (AC2) · 13 Dec 2016

**Reply to Reviewer 2**

We thank the reviewer for the critical thoughts on the manuscript and for raising important points which have led to changes to the manuscript, including additional Supplementary material.

A main line of criticism is that GlobSnow SWE cannot be a suitable data set for the purpose of ground thermal modeling in the Lena River Delta, based on the characteristics of our study area and the few published GlobSnow validation studies. The critical issue is how the studies mentioned by the reviewer can be related to the conditions in the Lena River Delta. Before we provide a point-by-point reply to the review, we would like to clearly state our assessment of these studies and evaluate their suitability for characterizing the performance of GlobSow for our study area (the Lena River Delta) and study period (2000-2014 with particular emphasis on the period 2002-2014 for which validation data for ground temperatures are available).

a)      **Luojus et al., 2010 and Takala et al., 2011**: Both studies present validation information for Eurasia using the INTAS-SCCONE data set (Kitav et al., 2002). The comparison to this data set represents (to our best knowledge) the only systematic GlobSnow validation that includes our study area in a geographical sense. However, Fig. 2 in Takala et al., 2013, suggests that only a small fraction of the data set is obtained from Arctic tundra sites (with a rough count, we came to <50/1264 sites, i.e. <5%), and even fewer (<15/1264, i.e. <1.5%) are located in Northeast Siberian lowland tundra with climate and landscape characteristics that are at least somewhat similar to the Lena River Delta. The overwhelming majority of the sites, on the other hand, is located in the Boreal Forest and in steppe environments, where completely different snow pack properties must be assumed (Takala et al, 2011, Derksen et al., 2012). We emphasize that the comparison to the INTAS-SCCONE data is a highly meaningful benchmark on a continental scale, at least for SSM/I-based GlobSnow SWE retrievals. However, the study only presents an evaluation of the entire data set und information on subregions is not given. Therefore, we cannot see why the results, especially the absolute values for RMSE and bias presented in Table 2 (Takala et al., 2011), should be valid also for small subregions like the Lena River Delta, in particular when shallow Arctic tundra snow is strongly underrepresented in the data set. Furthermore, it should be noted that the INTAS-SCCONE data set covers the years 1978-2000, for which the GlobSnow data are compiled with older generations of passive microwave sensors compared to our study period.

An important point is that SWE in the Lena River Delta is generally low enough so that saturation effects do not influence SWE retrievals: starting at about 120-150mm average SWE (Fig.  1 in Luojus et al., 2010), the GlobSnow retrievals become biased to too low values, as brightness temperatures are affected by radiation emitted within the snow pack (and not only by scattering of radiation emitted from the ground below). This effect is a consequence of the physics of snow grain - radiation interactions and the results can therefore be expected valid for all microwave remote sensors. For SWE values of 60mm and less (as in the LRD, Fig. 5f, our

study), the relationship between measurements and GlobSnow retrievals is on average linear. Therefore, the retrievals must be considered more reliable in this range, although GlobSnow to a certain extent overestimates measured SWE values (Takala et al., 2011). However, these low SWE values in the data set (roughly 30% is SWE<50mm, Fig. 7b, Takala et al., 2011) could in many cases represent data from areas other than Arctic tundra, so that the bias is once again difficult to interpret with respect to our study area.

b)        **Takala et al., 2011:** In addition to the INTAS-SCCONE data, Takala et al., 2011, present validation for Finland and Canada. The data set from Finland is from a region with a high density of ground stations so that an improved performance can be expected compared to regions without a dense station network. For this reason, a comparison is challenging due to the lack of ground stations in the vicinity of the Lena River Delta. However, the saturation effect at higher SWE values is again evident, confirming this feature also for areas with high density of ground stations. Furthermore, a comparison to extensive in-situ data sets from Northern Canada is presented. These data were mainly obtained during a snowmobile traverse in 2007 (Derksen et al., 2009), largely in tundra areas at latitudes between approx. 64 and 68° N which is an environment in many aspects comparable to our study area (located approx 72-74° N). Only average results for the entire data set are presented in Table 3 (which the reviewer explicitly refers to), showing a mean SWE of 120mm, an RMSE of 47mm and a mean negative bias of 36mm. Takala et al., 2011, comment on this: "The relatively high uncertainty over tundra regions (Table 3) is likely driven by three issues: the extremely sparse network of surface climate stations across the Canadian sub-Arctic, the complex microwave emission from lake rich snow covered tundra (see Derksen et al., 2009), and the extremely heterogeneous tundra snow cover which complicates the determination of 'ground-truth' SWE at coarse spatial resolutions."

The first point "sparse station network" is at least partly true also for the Lena River Delta, but we note that the WMO station in Tiksi is in a distance of 50 (eastern part) to 200 km (western part), much closer than the closest station for a large part of the N Canadian traverse (Fig. 5a, Takala et al, 2011). Furthermore, the environmental and climatic conditions at Tiksi are similar to the Lena River Delta, so that the snow pack properties inferred for Tiksi are very likely a good representation for the Lena River Delta, which is favorable for SWE retrievals in our study area (see Derksen et al., 2012 for a discussion of the role of snow pack properties in SWE retrievals). Also the second point, "high water body fraction", is clearly applicable to the Lena River Delta, as pointed out by the reviewer. Using Landsat (Schneider et al., 2009) and MODIS (MODIS water mask) based land cover classifications, we estimate the water fraction in the interior of the Lena River Delta (the part for which the modeling was performed) between 12 and 30% in 25km EASE grid cells, with a single grid cell in the Eastern Delta reaching 37% (of which more than half is estimated to be river arms, see below). Almost three quarters of the grid cells feature water fractions of less than 20%. However, the character of the water bodies is very different to the ones in N Canada: themokarst lakes and river arms dominate in our study area, while the traverse crossed the Canadian Shield where lakes are mostly a result of glacial erosion. The track of the traverse provided in Derksen et al., 2009, suggests that unforested, ground-ice-rich, flat lowland

areas only made up an insignificant portion of the traverse: most of the area E of Great Bear Lake is characterized as "continuous permafrost with low ground ice content and thin overburden or exposed bedrock" in the IPA permafrost and ground ice map. Hereby, an important difference regarding SWE retrivals with passive microwave could be that many of the thermokarst lakes in the Lena River Delta are shallow and can even freeze to the bottom in winter (Schwamborn et al., 2012, Antonova et al., 2016), while the Canadian lakes are in general deeper. With respect to microwave emission, this could be an important difference: microwave emissions become more similar to land areas, although the emission characteristics of fully frozen water bodies are not yet entirely clarified (Gunn et al., 2011).

Furthermore, the Lena River features a very low winter discharge, as much of the catchment is in the continuous permafrost zone. Despite of recent increases, we estimate the winter discharge to be only about 10% of the average summer discharge (Fig. 2 in Yang et al., 2002), and large river areas visible as water in summer-derived satellite imagery (which are hence classified as water in the above mentioned classifications) fall dry in winter, which will decrease the water fraction in particular in the central and eastern part of the delta considerably. Furthermore, shallow river arms will freeze to the bottom, similar as the above mentioned thermokarst lakes, so that we expect the true "open water" fraction relevant for microwave brightness temperatures in winter to be significantly lower than the open water fractions obtained from summer imagery (see above) suggest. As a consequence, we argue, that the high summer water fraction in some parts of the LRD (although it may affect passive microwave retrievals in particular in fall) is not a priori an exclusion criterion for GlobSnow in the LRD, especially since the comparison with in-situ data set in the relatively water-body-rich area around Samoylov Island suggests a satisfactory performance.

Third, the pronounced spatial variability at scales smaller than 25km is certainly an issue also in the Lena River Delta, although the landscape is generally flat (compare the images of the borehole sites in the Supplement to the revised version of the manuscript) in most parts and large snow drifts only occur in localized spots, such as edges of islands or thermokarst gullies, which we do not target with our modeling. For the extensive N Canada data set, spatial variability at small scales will only affect the RMSE and not the bias when comparing GlobSnow retrievals to small-scale in-situ data. Fig. 10 in Takala et al., 2011, is clear evidence that the spatial variability is extreme at least in some parts of the Canadian study area, with a spread from 40 to more than 200mm SWE. Such strong differences at scales of less than 25km could to a large part explain the high RMSE value found in the N Canadian data set. However, we find it encouraging with respect to our study that GlobSnow in the presented cases can indeed capture a SWE value in the center of the SWE distribution (Fig. 10, Takala et al., 2011), which would be a satisfactory representation for the average snow conditions in the grid cell.

Finally, the largest difference between the N Canada data set and the Lena River Delta is the strong difference in the absolute values of SWE, with on average 120mm instead of 40-60mm. For an average of 120mm and values of more than 200mm occurring regularly (Fig. 10), the considerable negative bias of 36 mm is an indication that the GlobSnow SWE retrievals could

partly be affected by saturation effects (see above) in this area, which is highly unlikely for the much lower SWE values in the Lena River Delta. Furthermore, we do not see why absolute uncertainties (both bias and RMSE) from the N Canada study could simply be assigned to the Lena River Delta (as suggested by the reviewer), despite the large differences in absolute values for SWE itself. As evident for Fig. 1 in Luojus et al., 2010, higher SWE values are associated with higher absolute uncertainties, while lower SWE values have a lower absolute uncertainty. We therefore argue that, if at all, relative errors from the N Canada data set should be employed when assessing the possible performance of GlobSnow SWE in the Lena River Delta. In the revised version, we show that the uncertainties found for the N Canda data set can then be reconciled with the comparison for Samoylov Island (see below for details).

c)     **Derksen et al., 2012:** This study is not a validation of the operational GlobSnow SWE product itself, but investigates an important aspect of the retrieval algorithm: it evaluates the landcover dependence of microwave emission, distinguishing between open tundra areas, forest and lakes. We have carefully evaluated the information provided in the publication with respect to the conditions in the LRD.

The study area of Derksen et al., 2012, is located at 58-59° N near Churchill, Canada. The mean annual air temperature in the Churchill area is approx. -6.5°C (Fig. 3a, Zhang et al., 2012), while it is around -12.5°C for Samoylov Island in the Lena River Delta (Boike et al., 2013), a significant difference, which most certainly causes differences in the freezing behavior of lakes and possibly the snow pack properties. Moreover, the radiation regime during winter is necessarily different due the latitudinal difference, with polar night conditions dominating in the Lena River Delta, while Churchill is located several 100km south of the polar circle. This factor may influence the snow pack properties and lake freezing. No information on the lakes (depth, origin) is provided, which makes it difficult to compare to thermokarst lakes in our study area.

In-situ observations of passive microwave brightness temperatures and snow pack properties were conducted for different landcover types at sites located approx. 5-15km from the coast of Hudson Bay (Fig. 1, Derksen et al., 2012), showing a strong landcover dependence. We note that the 25km EASE grid cell #2 (Fig. 1, Derksen et al., 2012), in which all in-situ measurements were conducted, is located directly at the coast (with even a small ocean fraction included), and the "ocean overspill problem" (see Other remarks by the reviewer, point 1) is not mentioned or taken into account in this study. However, only grid cell #2 facilitates, in our opinion, a direct comparison between in-situ and satellite brightness temperature measurements: the "ground truth information" for the other grid cells is synthesized from a landcover classification, assuming that the in-situ measurements conducted at localized sites in the vicinity of the coast can deliver unique values for "lake snow depth", "open tundra snow depth" and "forest snow depth" (and for all the other snow pack and lake ice properties) that are representative also for sites more than 50km inland. In this procedure, possible spatial differences in precipitation, temperature and wind speed (controlling snow redistribution) between coast and inland are not mentioned or taken into

account (although they may well exist, see Fig. 3a/b in Zhang et al., 2012 for temperature and precipitation).

Summarizing our assessment, we conclude that a) the two study areas feature somewhat different characteristics with respect to landcover and climate, and it is not entirely clear in how far this affects the transferability of the findings; b) no in-situ measurements are available from the grid cells other than one coast grid cell. The ground truth information is instead based on land-cover-weighted average of in-situ measurements taken near the coast.

In this light, we further evaluate Table 9 and Fig. 12 in Derksen et al., 2012, which are mentioned by the reviewer. Table 9 and 10 display absolute RMSE and bias for SWE retrievals for all grid cells, i.e. the data set affected by issue b) mentioned above. RMSE values strongly increase if grid cells other than # 2 (where the in-situ measurements were conducted) are evaluated, while there is only little effect of different lake fractions (<25% and <50% are distinguished with grid cell maximum lake fraction 73%, Table 3, note that the overwhelming majority of grid cells in the Lena River Delta would fall in the low lake fraction category). In our opinion, it can at least not be ruled out that the results in Tables 9 and 10 are affected by systematic differences in snow cover between grid cells on a coast-inland gradient that are not captured by the study design. Such differences could at least contrbute to the RMSE increase for grid cells other than #2. In addition, the results are strongly affected by the presence of forest as a third landcover class, featuring completely different snow depths and densities (Fig. 3, Derksen et al., 2012), which is not an issue in the Lena River Delta. This effect might even override the uncertainty caused by water bodies, at least for grid cells with lake fractions <25%. Finally, it is not clear what the results exactly mean for operational GlobSnow SWE retrievals, which are also controlled by the quality of the station-interpolated background fields. We conclude by noting that Derksen et al., 2012, do not share the reviewer's pessimistic view with respect to passive microwave retrievals. They write: "The results in Table 9 are encouraging with respect to passive microwave SWE retrievals, however, this represents an ideal scenario in which snow cover characteristics were thoroughly measured through the complete winter season and available for input to the forward modeling component of the retrieval. In order to test retrieval performance under less idealized circumstances, the retrieval simulations were re-run with only a single land use tile. This better replicates an operational scenario where information from only a single snow survey or weather station would be available. Because these observations are often located in open areas (i.e. adjacent to airports) only the snow measurements from the open site were used as model inputs. Table 10 provides a summary of these retrievals; the accuracy is not influenced appreciably (and actually improved in some cases) compared to the simulations using the full set of snow observations. This suggests that snow information for a single land cover class can still result in useful retrievals. "

Fig. 12 in Derksen et al., 2012, shows the spatial variability of SWE within and between landcover classes (based on the in-situ measurements in grid cell #2), in conjunction with the single value retrieved by the satellite, which we presume corresponds to grid cell #2, containing 62% open, 27% forest and 11% lake (Table 3). When roughly computing the grid cell average

with the above landcover fractions, we find that the satellite retrievals seem to match quite well, at maximum overestimating the in-situ value with about 25-30% and thus far from the 100% error mentioned by the reviewer. Moreover, presence of forest significantly complicates the picture compared to the Lena River Delta. Considering the high spatial variability within and between the classes and the fact that systematic sampling over the entire 25km grid cell has not been performed, it is, in our opinion, not entirely clear if this moderate mismatch is due to the scaling of the in-situ data or the satellite retrievals. The obvious decrease in satellite-derived SWE in March (that clearly must be associated with a higher uncertainty) might be explained by formation of ice lenses, as described for March in the paper (an effect that generally does not occur in the colder winter climate of the LRD). As a consequence, we do not agree with the reviewer's statement regarding Fig. 12 ("even more revealing (over 100% error; Figure 12) when examining errors on a monthly basis"), at least when regarding grid cell average SWE. It is clear that mismatches of more than 100% occur for point measurements due to the strong spatial variability of SWE, but this is an inherent issue in coarse-scale products such as GlobSnow SWE, which aim at delivering grid cell averages.

In the following, we provide point-by-point replies to all issues raised by the reviewer. In the end, we summarize the major changes to the manuscript that were inserted following the review. The reviewer's comments are provided in bold font, our replies in normal font and changes to the manuscript in italics:

**This paper presents an approach to map the spatial distribution of ground temperatures and thaw depths using a 1D transient ground thermal model (CryoGrid 2). The model uses remote sensing derived surface temperature (MODIS 1 km complemented by 2-m air temperature from atmospheric reanalysis ERA-Interim 0.75 deg. grid spacing) and snow depth obtained from the GlobSnow snow water equivalent (SWE; 25 km grid spacing) product as forcing data. The study builds on the earlier work of Langer et al. (2013), moving the application of the CryoGrid 2 model from the local scale (station on Samoylov Island) to the regional scale by including the entire Lena River Delta (LRD). From Figure 1, one notices that the LRD is covered by many small lakes and branches of the Lena River (i.e. a large freshwater fraction).**

We agree. We now provide quantitative information of the water fractions of the GlobSnow grid cells in the Lena River Delta. More than 70% of the employed grid cells have open water fractions of less than 20%, and only a single grid cell has a high fraction of more than 35%, most of which are river arms. Grid cells near the coast with even higher open water fractions do not contribute to our model forcing data set, these areas have been excluded. As detailed in the above assessment of Takala et al., 2011, these open water fraction correspond to the summer state. As

the winter discharge of the Lena River is reduced to approx. 10% of the summer flow (Yang et al., 2002), large river areas must be expected to fall dry during winter. Furthermore, many shallow water bodies freeze to the bottom in winter in the Lena River Delta (see Schwamborn et al., 2002, and Anotonova et al., 2016, for field data), so that significantly lower "winter open water fraction" can be expected compared to the summer values, at least after mid winter. Therefore, we do not see a reason why the performance of GlobSnow should be significantly worse than in lake-rich areas in N Canada (Takala et al., 2011).

**This is a complex area to study using coarse resolution passive microwave satellite data (or derived products) due to the large sub-grid scale variability within pixels,**

Regarding the spatial variability of SWE, our scheme does not aim for directly capturing the small-scale variability of snow depths within 1km grid cells, but a significant scale mismatch remains between the 25km GlobSnow pixels and the 1km model resolution (which we discuss in detail in Sect. 5.2), although it is unlikely to assume that abrupt changes in average SWE occur in the flat landscape of the LRD. Nevertheless, this mismatch cannot be resolved, and is therefore likely to constitute an important source of uncertainty. However, when we compare the model output to point measurements of the ground thermal regime, we find it encouraging that the model results in most cases can fit the measurements, which could be considered an indication that both model and measurements reproduce "average conditions" in the relatively flat tundra landscape of the Lena River Delta. In the revised version, we have added Supplementary material showing images of the borehole sites, which demonstrates that the landscape is indeed rather homogeneous in the vicinity of the boreholes.

**notably due to the presence of water bodies, which introduces significant uncertainty in SWE estimates (GlobSnow or other satellite-based SWE products).**

We partly agree. It is evident that water bodies have an effect on the brightness temperatures obtained by passive microwave sensors, leading to problems in SWE products based on passive microwave data. However, the GlobSnow SWE retrieval features a data assimilation procedure, which also takes in-situ measurements at WMO stations into account. The station data are interpolated in space to provide a background or "a priori" field for the assimilation of the satellite data. This background field is then weighted against SWE information derived from the passive microwave sensor. Takala et al., 2011, state: "A basic feature of the algorithm is that if the sensitivity of space-borne radiometer observations to SWE is assessed to be close to zero (…), the weight of the radiometer measurements on producing the 'assimilated SWE' approaches zero (…). The higher the estimated sensitivity of TB to SWE, the higher the weight given to the radiometer data. Thus, the weight of the radiometer data varies both temporally and spatially in order to provide a maximum likelihood estimate of SWE." It is beyond the scope of our study to evaluate in detail how GlobSnow retrievals are obtained in the Lena River Delta. However, it is clear that the GlobSnow algorithm is capable of handling situations with reduced reliability of the passive microwave retrievals, other than algorithms entirely based on satellite retrievals.

Therefore, studies focusing solely on the effect of lakes on passive microwave brightness temperatures provide an incomplete picture on how the GlobSnow algorithm will handle these situations.

**This issue has been recognized by the group who originally developed the GlobSnow product (reported in Takala et al., 2011) and at least one of its members in a latter publication (Derksen et al., 2012). Takala et al. (2011; Table 3) report mean bias errors of -36 mm and RMSE of 47 mm for a tundra area with small water bodies.**

Table 3 in Takala et al. (2011) is also clear evidence that the absolute values of SWE in the Canadian data set are considerably higher than in our study area, 120mm vs. 35-60mm. Therefore, we do not find it adequate to assign absolute errors from this data set to the much lower SWE values in the Lena River Delta (see our detailed assessment above). In fact, if one simply did that without further thinking, negative SWE values would occur regularly in the Lena Delta, which is impossible both in reality and in the GlobSnow processing algorithm. Instead, if we do assume that error estimates can be transferred from the N Canada data set to the Lena River Delta, we suggest that relative errors are more appropriate.

On Samoylov, we have an average snow depth of ca. 16 cm at $225kg/m^3$ (Fig. 5f) which corresponds to a SWE of about 40mm. Assuming a relative error similar to the Canadian study (RMSE of 47mm at 120mm average SWE, i.e. a 40% relative error), we obtain an absolute error of 15mm SWE (i.e. 6-7cm snow) at Samoylov. Considering our comparison of snow depth model forcing to in-situ data (Fig. 3, our study), this RMSE appears indeed realistic: using only the non-zero in-situ snow depth measurements, we obtain an RMSE of 6 cm for the snow depths, while the average bias of +1.5cm is of opposite sign and considerably smaller than for the N Canadian data set (-36mm at 120mm total SWE corresponds to ca. -5 cm snow depth, if scaled to the average SWE values from Samoylov Island). This underestimation in the N Canada data set could at least partly be due to saturation effects at SWE values >150mm, which is not an issue in the LRD. Although only a single point measurement is available, this comparison is an indication that the performance of GlobSnow-derived SWE in the LRD is at least not worse than in other arctic areas.

**Derksen et al. (2012) also show that SWE retrieval errors can be large (see Table 9 and Figure 12 of this publication) from passive microwave data, even more revealing (over 100% error; Figure 12) when examining errors on a monthly basis.**

Please see our detailed discussion of Derksen et al. (2012) above. The study showcases the land-cover-dependence of snow depth/properties and the effect on passive microwave emissions, but we do not think that the findings can be regarded as evidence that GlobSnow is entirely inadequate for the LRD. If we re-perform the above scaling exercise for the values given in Table 9 and Fig. 12, we once again arrive at an order of magnitude for the uncertainty that is well comparable to the performance for Samoylov Island (see previous point).

**As indicated by Takala et al. (2011): "Additionally, the consideration or compensation for the effect of (frozen) lakes requires further study and algorithm development work."**

Our model scheme is designed in a way that improved future SWE products can be directly ingested, which consequently has the potential to improve the modeled ground thermal regime. In the revised version of the manuscript we write in Sect. 5.1: "*In the future, enhanced SWE retrieval algorithms taking waster bodies explicitly into account (e.g. Lemmetyinen et al., 2011) may become available.*"

**The authors of the present manuscript state, in Langer et al. (2013), that: "The thermal state of permafrost is reproduced with an uncertainty of about ±2.5 °C with a SWE accuracy of about ±10 mm.**

This statement is based on a sensitivity analysis at a single point which applies a constant bias to the entire time series. In this study, however, we provide a characterization of uncertainties based on a comparison to in-situ measurements. As the number of validation sites is limited, we have added the following statement to the Conclusion of the revised version of the manuscript: "*However, due to the relatively small sample of validation sites, this accuracy assessment must be considered preliminary.* "

**This is still below the performance that can be reached with a realistic LST accuracy of about ±2 °C. However, a much lower SWE accuracy level (±40 mm) must be considered in regions with sparse weather stations (Luojus et al., 2010) and when field measurements are not available for calibration. Our results show that realistic permafrost simulations with a transient heat transfer model would be almost impossible with such low accuracies in the SWE forcing.**

Here, Langer et al., 2013, refers to the accuracy stated in Luojus et al., 2010, which is once again intended as a global benchmark, including many different conditions (see above). It is not intended as a benchmark on the regional scale. In the context of this statement, it represents a worst case that one can possibly encounter if the model scheme of Langer et al., 2013, (with model parameters of Samoylov Island) is applied to a random point in the Northern Hemisphere. However, in the present study we are interested in the performance in the Lena River Delta itself, and the above considerations are clear indication that an absolute error of 40mm is not adequate.

**In contrast to the permafrost temperatures, the thaw depths are found to be more or less independent from the SWE accuracy. However, this might be different in regions where the permafrost temperature is already close to the freezing point as observed by Åkerman & Johansson (2008). In any case, the impact of snow on the active layer dynamics can be very complex and dependent on regional factors (Zhang, 2005). The performed sensitivity study demonstrates that a highly accurate snow cover forcing is crucial for reliable permafrost modeling."**

This statement once again refers to application at a random point. From Langer et al., 2013, it is not really clear what "highly accurate snow cover forcing" means, and it is necessary to evaluate this point in further regional studies – this is exactly our intention with the present manuscript, and we will evaluate the scheme further in other case studies in the future.

**Given: 1) the above statement by the authors in a previous paper;**

As detailed above, the present study is a follow-up of Langer et al., 2013, and in many aspects represents an update. Some deviations in the findings are therefore not surprising, although we argue that the SWE threshold estimates can largely be reconciled with GlobSnow error estimates from Takala et al., 2011, if relative instead of absolute errors are employed. The statements from Langer et al., 2013, as picked by the reviewer, clearly refer to application at a random point anywhere in the world, not to the relatively similar landscape of the Lena River Delta, which even includes the study site of Langer et al., 2013.

**2) the known retrieval errors in similar regions reported by the developers of GlobSnow SWE;**

As detailed above, we do not consider absolute errors derived from continental-scale studies applicable for the LRD. If they were, the satisfactory agreement at Samoylov Island (Fig. 3, our study) could only be explained by fortuitous coincidence. If we assume relative errors instead, the reported errors from "similar" regions can largely be reconciled with our comparison for Samoylov Island.

**and 3) the lack of validation of snow depth (derived from GlobSnow SWE with density values of 200-250 kg m-3 ) over a larger area (transects) than just the small island of Samoylov (located to the south of the LRD),**

This relates to "Other remarks, #4: **How much confidence should we have in the snow depth map of Figure 5"**? We agree, this is a very relevant question, in particular whether the slight increase of snow depths/SWE from E to W is real. In the revised version of the manuscript, we provide Supplementary Material in which we compare our GlobSnow-based forcing data with Canadian Meteorological Centre (CMC) Snow Depth Analysis Data (Brasnett, 1999; Brown & Brasnett, 2015). The CMC product provides SWE values at a spatial resolution of 24km, comparable to GlobSnow SWE, and has been used as a reference product to evaluate global snow retrievals (Frei et al., 2012). The CMC product does not employ satellite-derived passive microwave data for deriving SWE, and is hence completely unaffected by water bodies. Instead, a background field of SWE is calculated from snowfall in an atmospheric circulation model, which is subsequently updated by assimilating in-situ measurements from WMO ground stations. Both data sets use WMO data from Tiksi which could affect the absolute values, but not the spatial patterns, as no other WMO station is located close-by to the W or N of the Lena River Delta. For the comparison, we have used the CMC monthly SWE data set for the period 2004 to

2013, corresponding to the period displayed in Fig. 5 of our manuscript. The result of the spatial comparison is shown in Fig. R1. Both products show a similar spatial pattern and absolute values mostly agree to within 10mm, with CMC generally showing lower values than GlobSnow. When interpolated to 1km scale, a significant correlation between the data sets is found ($r^2$=0.71). The coarse-scale pattern in the LRD is further backed up by winter precipitation from the ERA-interim reanalysis (Fig. R2). Despite of the coarse resolution and the insufficient representation of the coastline, there is a clear W-E gradient over land areas in the LRD, with lowest values occurring in the SE edge, similar to GlobSnow SWE.

While these comparisons to independent model data sets do not constitute a validation of GlobSnow SWE in a strict sense, they indicate that the large-scale pattern derived from GlobSnow is not an artefact of the GlobSnow SWE retrieval, but related due to regional trends in winter precipitation.

[Figure]

Fig. R1: Average SWE in the months October to June from 2004 to 2013. Left: Canadian Meteorological Centre (CMC) Snow Depth Analysis Data, 24 km resolution; the black line corresponds to the outline of our model domain. Right: CryoGrid 2 forcing data (1km resolution) based on GlobSnow SWE and MODIS snow cover. Note the offset of 5 mm between the color scales.

[Figure]

Fig. R2: Annual averages of the total precipitation ([mm]) falling at 2m-air temperatures of less than 0°C for the months October to June for 2004-2013, based on the ERA-interim reanalysis at 0.75° resolution (values represented as grid cells). The land-sea mask is indicated by a white line.

**I am afraid to say that the manuscript submitted is not acceptable for publication in The Cryosphere. In fact, I am quite concerned by the fact that the authors missed the publication of Takala et al. (2011) which is the key paper reporting uncertainties of the GlobSnow SWE product. It is important to read and cite others who work in similar areas or at least with similar data sets, and who have reported uncertainties in the forcing variables used by CryoGrid 2.**

We have provided a detailed assessment of Takala et al., 2011, and further literature on the GlobSnow algorithm (see Major changes to the manuscript). We point out that Takala et al., 2013, is clearly dedicated to a continental-scale assessment, and caution is warranted when assigning the error estimates to regional scales. We believe that absolute error values taken from this study are inappropriate for the Lena River Delta (see above).

**Other remarks:**

**1. The authors do not seem to be aware that the SSM/I footprints for the 19 GHz and 37 GHz frequency brightness temperature channels are in the order of 70x45 km and 38x30 km, respectively. These brightness temperature measurements are then interpolated into a 25x25 km grid which is then used for SWE retrieval in GlobSnow. Therefore, although the authors masked some areas along the coast, the ocean "overspill" problem within the footprints is a larger problem than reported herein.**

We have generally kept a distance of 20-30km from the coast, so that contributions from the ocean are at least not large, considering the footprint sizes. At the western end of the modeled domain, the distance to the coast is less in order to be able to include the validation sites "Olenyoskaya channel mouth" and "Turakh Island". We have stated in the revised version that a higher uncertainty is likely for these sites.

**2. The large fraction of the landscape covered by lakes/river channels represents the largest uncertainty in GlobSnow SWE values. The authors need to read further on this topic in order to better understand the limitations of GlobSnow SWE and, perhaps, search for other products (satellite or reanalysis, including assemble) that could be considered in a new manuscript submission to The Cryosphere or another journal.**

Please see our detailed assessment above.

**3. Boike et al. (2013) is given as the reference for the snow depth and density values of Samoylov Island. However, I personally browsed this paper to find that there are mismatches between values reported in Table 5 and Figure 6 of that paper and the values reported in Figure 3 (and the text) of the present manuscript. I am not sure how, as a reviewer, I can reconcile the two sources. The range (and maximum) of measured snow depths in Boike et al. (2013) do not always match those of this paper. For example, in winter 2004 (a high snow year), the maximum snow depth found in Table 5 and Figure 6 of Boike et al. (2013) is 56 cm while that plotted in the graph of Figure 3 of this paper is at a value of about 47 cm. This is only one of several examples.**

In Fig. 3, we have applied a running average filter with a weekly window, corresponding to the temporal resolution of our forcing data. This is now stated in the figure caption of the revised version, and explains deviations from Table 5, where the absolute maximum recorded is displayed: differences of several centimeters can be easily explained by snowfall during periods of low wind speeds that gets quickly removed or compressed by wind action afterwards. Fig. 6 in Boike et al., 2013, refers to a different data set, i.e. spatially distributed measurements taken at one point in time. We have now added this spatial survey as a data point to Fig. 3 in the revised version manuscript, which indicates that GlobSnow SWE is an adequate representation.

**4. How much confidence should we have in the snow depth map of Figure 5 and the ground temperature (1-m depth) map of Figure 11, given that snow density comes from Samoylov Island only and that there is a large degree of uncertainty in GlobSnow SWE retrievals over complex (lake-rich) areas such as the LRD?**
We have provided a comparison to an independent SWE data (Figs. R1, R2), confirming the SWE general pattern in the Lena River Delta. Please see our detailed response above.

**As shown in Figures 6-8, winter temperatures are significantly underestimated in wintertime by the model (up to 8°C, most frequently by 3-4°C). Of course, taking the**

**average of all years combined reduces the error reported (1-1.5°C given in the Abstract), but the errors are larger when inspecting each individual year.**

We do not agree with the reviewer's statement "winter temperatures are significantly underestimated in wintertime by the model (up to 8°C, most frequently by 3-4°C)". We presume that the reviewer refers to periods when a) thermokarst development at the borehole sites was obvious (Sardagk and Kurunghak), or b) a change in the snow regime due to the building of a new research station had taken place (Samoylov). The timing of these events was described rather qualitatively in the original version of the manuscript. In the revised version, we have now clearly marked the affected periods in Figs. 7 and 8 and stated in the figure captions that they should not be employed for comparison. We also provide images showing the thermokarst development around the boreholes (as well as new buildings around the borehole on Samoylov Island) in the new Supplement

To facilitate a more quantitative evaluation of model results for the annual cycle, we have added a new Fig. 9, which displays a scatter plot for monthly averages for all boreholes. In agreement with the comparison of yearly averages, we find that the model results have a slight cold-bias of - 0.9°C and an RMSE of 1.1°C (for a snow density of 225 kg/m$^3$).  In the revised version, we write in Sect. 4.2.1:

*"A comparison of monthly averages for all five boreholes is shown in Fig. 9. For a snow density of 225 kg m$^{-3}$, the model results feature an RMSE of 1.1°C and an average bias of -0.9°C, mainly due to underestimation of measured values during the summer and fall seasons. For a snow density of 200 kg m$^{-3}$, the model bias is on average positive (+0.8°C), but the RMSE is increased (1.6°C). The model performance is worst for the highest snow density (RMSE 2.1°C, bias -2.1°C). If the Samoylov Island borehole (for which the ground stratigraphy was adjusted, see above) is removed, the model performance for the best-fitting snow density of 225 kg m$^{-3}$ remains largely unchanged (RMSE 1.2°C, bias -0.9°C)."*

**5. The scaling issue between point (single station measurement(s)) and large satellite pixels should not be ignored throughout the manuscript.**

We agree, and we have mentioned the scale mismatch several times in the manuscript. Our in-situ measurements are generally located at points that the installation team deemed to be representative for the larger-scale environment, which may partly explain the satisfactory agreement with the model results. In the revised version, we provide Supplementary material with images of the borehole sites, which show the flat and relatively homogeneous landscape around the borehole sites.

**Major Changes to the manuscript in response to the reviewer:**

Sect. 3.3 Model forcing data:

[revised manuscript text omitted]

**Additional References:**

Antonova, S., Duguay, C., Kääb, A., Heim, B., Westermann, S., Langer, M., & Boike, J. (2016). Monitoring bedfast ice in lakes of the Lena River Delta using TerraSAR-X backscatter and coherence time series, Remote Sensing, 8(11), 903; doi:10.3390/rs8110903.

Brasnett, Bruce. 1999. A Global Analysis of Snow Depth for Numerical Weather Prediction. Journal of Applied Meteorology 38: 726–740.

Brown, Ross D. and Bruce Brasnett. 2015, updated annually. Canadian Meteorological Centre (CMC) Daily Snow Depth Analysis Data. © Environment Canada, 2010. Boulder, Colorado USA: National Snow and Ice Data Center.

Derksen, C., Silis, A., Sturm, M., Holmgren, J., Liston, G. E., Huntington, H., & Solie, D. (2009). Northwest Territories and Nunavut snow characteristics from a subarctic traverse: Implications for passive microwave remote sensing. Journal of Hydrometeorology, 10(2), 448-463.

Frei, A., Tedesco, M., Lee, S., Foster, J., Hall, D. K., Kelly, R., & Robinson, D. A. (2012). A review of global satellite-derived snow products. Advances in Space Research, 50(8), 1007-1029.

Gunn, G. E., Duguay, C. R., Derksen, C., Lemmetyinen, J., and Toose, P.: Evaluation of the HUT modified snow emission model over lake ice using airborne passive microwave measurements, Remote sensing of Environment, 115, 233–244, 2011.

Lemmetyinen, J., Kontu, A., Kärnä, J.-P., Vehviläinen, J., Takala, M., and Pulliainen, J.: Correcting for the influence of frozen lakes in satellite microwave radiometer observations through application of a microwave emission model, Remote Sensing of Environment, 115, 3695–3706, 2011.

Yang, D., D. L. Kane, L. Hinzman, X. Zhang, T. Zhang, & H. Ye (2002). Siberian Lena River hydrologic regime and recent change, J. Geophys. Res., 107(D23), 4694, doi:10.1029/2002JD002542.

Zhang, Y., Li, J., Wang, X., Chen, W., Sladen, W., Dyke, L., ... & Kowalchuk, S. (2012). Modelling and mapping permafrost at high spatial resolution in Wapusk National Park, Hudson Bay Lowlands, Canadian Journal of Earth Sciences, 49(8), 925-937.

---

## Author Response (AR2)

**Authors Response**

Two referee reports are available, one of which does not raise any new points. We provide point-by-point responses to the Editor comments, as well as the report by Referee 3. This is followed by a revised version of the manuscript, with changes marked in bold font.

*Editor comments:*

**Based on the comments from Anonymous Referee #3 and my reading of the manuscript, I am happy that this manuscript can go ahead towards publication. I ask you to address the comments made by the referee as well as my comments below and included in the annotated PDF I have provided.**

All annotations from the PDF have been incorporated in the revised version. Thank you very much for the detailed suggestions, and please find more detailed responses below!

**1) Temperatures can be high/low, but not cold/warm (objects are).**

We agree, this has been corrected throughout the manuscript.

**2) P18L21: Your snow density values were determined by in-situ measurements. Please mention this in Section 5.4 where you list challenges for extending this approach beyond the LRD. Not having these measurements may be difficult.**

The snow density is a crucial parameter in the model scheme which has been determined from in-situ measurements in this study. For application on larger domains, spatial differences in snow density must be considered, which might be obtained e.g. from simple empirical relationships with climate variables (Onuchin and Burenina, 1996)."

**3) In the last few sentences of your conclusion, please point to Section 5.4. I believe that identifying many of the challenges for extending your study past the LRD make a big part of the value of this study – and the reviews it received.**

A reference to Sect. 5.4 has been inserted.

**4) P20L7: "If this interpretation is correct, surface temperatures derived from remote sensors have a significant advantage over data sets derived from atmospheric modeling…". Even if this is not correct and pure coincidence, your statement on the value of remote sensing data would be true: they contain data that re-analyses do not currently resolve.**

We agree and have deleted "If this interpretation is correct"

**5) You often judge your model to be "adequate". This needs to (a) have a purpose declared for which it is adequate and (b) criteria explained that will allow you to judge the model to be either adequate or inadequate. Without these two additions, stating adequacy has no value.**

We agree. As it is cumbersome (and not always possible) to be explicit about both points in most of the instances where "adequate" was employed without additional specifications (as marked in the annotated manuscript), we have adapted the formulations individually.

**6) P14L7: Why is the accuracy defined with respect to only one snow density value if you cannot plausibly constrain the densities better than 200–250 kg/m3? It would be good to also give the pessimistic estimates derived from the extreme densities. Figure 9 is a great visualization of this.**

In the revised version, we provide an overall accuracy estimate of 1 to 2 °C for multi-annual averages, instead of 1 to 1.5 °C in the previous version (in abstract and conclusion). This includes the uncertainty due to the poorly constrained snow densities. We consider this adequate, considering Fig. 10 – there is only one case (high snow density for Sam) where the deviation is slighly larger than 2 °C. We explain this in more detail in Sect. 4.2.1: "For the average snow density of 225 kg m$^{-3}$, the measured and modeled values agree within 1 to 1.5°C, which can serve as a coarse accuracy estimate for the spatially distributed simulations of the ground thermal regime in the LRD (Fig. 12, see Sect. 4.2.2). If snow densities are allowed to vary between 200 and 250 kg m$^{-3}$, the agreement is generally better than 2°C.

*Referee 3:*

**This work updates a previous study by spatially distributing a point model scheme described in Langer et al (2013) across the Lena River Delta. The paper is clearly written and provides clear evidence of decent model performance despite the heterogeneous environment which complicates the characterization of snow cover (amongst other variables). The study clearly identifies the link between snow and the ground thermal regime, and the impacts of shortcomings in observed snow mass datasets. I suggest the following issues be addressed to improve the final manuscript:**

**Page 7 line 1: "CryoGrid 2 is capable of representing the annual build-up and disappearance of the snow cover with a variable number of snow grid cells…" Not clear what is meant by a 'variable number of grid cells'. Does the model not simply simulate snow accumulation/ablation on a grid cell by grid cell basis?**

Yes, that is true! We have changed the formulation to "CryoGrid 2 is capable of representing the annual build-up and disappearance of the snow cover by adding/subtracting grid cells…"

**Page 8 line 5: given the importance of snow to model performance of the ground thermal regime, some additional details on the snow component of CryoGrid 2 should be provided. Presumably the precipitation forcing is key to the simulation of a realistic snow cover? What snow processes are simulated by the model (i.e. sublimation loss during blow snow events is likely important in this environment)? It's not clear in Section 3.4 how layering in the snowpack is treated.**

CryoGrid 2 does not explicitly account for processes, such as snowfall, melting, sublimation, meltwater infiltration, etc. The model basically adds snow layers with different thermal properties on top of the ground layers, according to the time series of snow depth compiled from the satellite products. The thermal properties of the snow gird cells are determined from the (constant) snow density. We have made this clearer in the model description in Sect. 3.3: "CryoGrid 2 is capable of representing the annual build-up and disappearance of the snow cover by adding/subtracting grid cells according to a time series of snow water equivalent (which must be provided as part of the forcing data), …"

**Section 3.3 – I'm confused by the terminology in this section. To me, the MODIS LST product provides a skin temperature, while the reanalysis provides 2 m air temperature. Given the effort required to use the MODIS product, I assume it is skin temperature that the model requires for forcing, not surface air temperature? This also would explain the assumption that temperatures are fixed at 0 degrees when snow is present (which is reasonable for skin temperature, not for surface temperature). If I have indeed interpreted this correctly, please be consistent with the terminology in this section – for example (page 8 line 26) "For melting snow, surface temperatures…" should be changed to 'skin temperatures' etc.**

In the article, we have followed the naming convention chosen for the MODIS products, land "surface temperature", which is used equivalent to skin temperature. This is required to force the model. In the revised version, we have clarified this issue by changing the first sentence of Sect. 3.3 to: "CryoGrid 2 requires time series of surface (i.e. skin) temperatures and snow water equivalent as forcing data sets."

**GlobSnow SWE: The Lena River Delta is a very challenging environment for SWE retrieval given the high water fraction, organic soils, coastal location, etc. Given the distance to the nearest ground stations, I would expect a high level of uncertainty in the GlobSnow retrievals. On balance, however, I feel the authors provide a convincing case in the supplementary material that the pattern of SWE they derive from MODIS and GlobSnow is reasonable when compared with the CMC product and clear text is provided in numerous places throughout the paper on uncertainties in GlobSnow.**

**Section 4.1: I'm not that familiar with this product, but is a cold bias in the MODIS LST data consistent with other literature? (it appears so, as you cite some studies later on page 15; I suggest you also include these citations in Section 4.1)**

We are unsure about this comment, we have actually cited a number of studies on the cold-bias in Sect. 4.1. The study Liu et al., 2004, which is cited in addition on page 15 does not really fit to Sect. 4.1, as it mainly deals with the MODIS cloud detection algorithm, which we so not evaluate any further in Sect. 4.1.

**Page 11 line 24: note that the GlobSnow SWE retrieval uses a fixed snow density of 0.24. I assume this was used to back out snow depth from SWE for this product? If the assumption of density =0.24 is applied to GlobSnow to estimate depth, should that same value be used for simulations instead of 0.225? (Perhaps this small difference in density is not important, especially since 0.24 falls within the 0.2 to 0.25 range in density used for the simulations)**

This assumed snow density of 0.24 does not enter the GlobSnow SWE retrieval as a simple scaling parameter between SWE and snow depth, but is also employed to e.g. estimate effective snow grain size (a crucial parameter in the emission model) at the reference stations, which is then used to determine the background field of effective snow grain sizes required for SWE retrievals on the grid cell level (Takala et al., 2011). Therefore, the snow density of 0.24 has the character of a global model parameter in the retrieval and computing a "GlobSnow snow depth" by using it as a scaling parameter is questionable. We rather employ the final product, the GlobSnow SWE values, for which also the cited validation studies have been performed.

We have clarified the procedure in the mode description, Sect. 3.1 where we state that SWE is used as model forcing. "CryoGrid 2 is capable of representing the annual build-up and disappearance of the snow cover by adding/subtracting grid cells according to a time series of snow water equivalent, (…) Furthermore, the snow density is employed to compute the volumetric heat capacity of the snow and to convert snow water equivalent to snow depth."

Furthermore, we have inserted a statement in Sect. 3.3, where we discuss GlobSnow retrievals in the LRD: "In the data assimilation procedure, a spatially constant snow density of $240 kg/m^3$ is assumed, which is in the range of the in-situ measurements on Samoylov island (Sect.3.2).

**Figure 5 panel f: This panel essentially shows uniform snow depth across the domain because only 5 cm is covered by the 4 classes in the legend. Based on the spatial survey shown in Figure 4, this spatial variability is dwarfed by the sub-grid heterogeneity. Presumably the seasonal averaging smooths some of the spatial distribution, but based on the variability in Figure 4 and the pattern in Figure 5, one could argue that the 1 km resolution average snow depth is essentially the same across the domain despite a high degree of sub-grid variability. How sensitive are the ground thermal regime simulations (Figures 12 and 13 for instance) to the use of spatially varying snow depth versus fixed snow depth?**

We agree, the spatial differences are relatively small (similar to what the CMC and ERA reanalysis products show), and the main point for the model fit is that the general magnitude of the snow depths is reproduced. However, the maximum relative differences in the LRD are still in the range of 25% which is not entirely insignificant. The general sensitivity of model results towards SWE has been evaluated by Langer et al. (2013) and 10mm SWE (which is approximately the maximum difference in the LRD, Fig. 5f) roughly correspond to a difference in 2.5 °C at 2.5 m depth, which gives an impression of the impact of the SWE pattern on Fig. 12. For thaw depths (Fig. 13), the impact is rather limited (see also Langer et

al., 2013).We do not think that runs with spatially fixed snow depth can add to this analysis, as it would require selecting and describing a reference time series (e.g. an average of all grid cells, or gap-filled in-situ measurements) and motivate its choice, which would make the manuscript even more diverse and less focused. Concerning the spatial variability superimposed on the larger-scale patterns, this is mentioned several times throughout the manuscript and discussed in detail in Sect. 5.2.

**Section 5.1.2: This is a very thorough discussion of uncertainties in SWE retrievals. One issue:**

**Page 16 lines 13-15: The suggestion here is that a temporally continuous 10 mm SWE error results in a 2.5 degree error in temperature at 2.5 m depth. This is a sobering number (analogous to the issue in altimetric retrievals of sea ice thickness where errors in snow depth propagate to larger errors in freeboard/ice thickness retrievals). The conclusion is that SWE errors of this magnitude are unlikely in this study because ground temperature simulations had smaller errors than 2.5 degrees. This ignores the effect of compensating errors (i.e. SWE biased too high and too low during various times of the season; impact of errors in ground stratigraphy; biases in temperature forcing etc.) and I think it's actually quite unlikely that SWE biases from GlobSnow were <10 mm due to all the uncertainties provided in the manuscript. I suggest the wording at the end of this paragraph be clarified just to be clear that uncertainty in the characterization of snow cannot be directly inferred from the overall simulation uncertainty.**

We agree, this paragraph represents an effort show the consistency of the different studies, results and data sets, in order to answer the criticism raised by Reviewer 2. However, it does not really fit in to the rest of the discussion point, and as the reviewer points out, the conclusion reaches too far. We have therefore deleted the corresponding passage completely in the revised version, which we think is adequate, as it also reduced the considerable length of Sect. 5.1.2 a bit. The first bullet point in Sect. 5.1.2 now finishes with: "Although the character of the two data sets differs (spatial transect vs. multi-year point measurement), the good agreement is an indication that the GlobSnow performance in the LRD could be similar to N Canada. We emphasize that the RMSE corresponds to undirected fluctuations around the average value which have much less influence on the modeled average ground thermal regime 5 (Figs. 12, 13) than a systematic bias."

**General comment on the description of the model - various terms are used such as a "satellite-based model scheme" (Abstract line 20) versus "…transient ground temperature modeling scheme forced by remote sensing data…" (Page 2 line 26). I think the second is more accurate as the model itself is not based on remote sensing, but remotely sensed data comprise most (but not all) of the forcing data.**

We fully agree! "satellite-based model scheme" has been replaced in both abstract and the final section of the Conclusion.

[revised manuscript text omitted]